

# An integrated observation dataset of the hydrological-thermal-deformation dynamics in the permafrost slopes and engineering infrastructure in the Qinghai-Tibet Engineering Corridor

Lihui Luo [1, 2, 3], Yanli Zhuang [1, 3], Mingyi Zhang [1, 2, 3], Zhongqiong Zhang [2, 3], Wei Ma [2, 3], Wenzhi Zhao [1, 3], Lin Zhao [4], Li Wang [5], Yanmei Shi [6], Ze Zhang [2, 3], Quntao Duan [1, 3], Deyu Tian [1], Qingguo Zhou [7]

[1]Northwest Institute of Eco-Environment and Resources, Chinese Academy of Sciences, Lanzhou 730000, China
[2]State Key Laboratory of Frozen Soils Engineering, Northwest Institute of Eco-Environment and Resources, Chinese Academy of Sciences, Lanzhou 730000, China
[3]University of Chinese Academy of Sciences, Beijing 100049, China
[4]Cryosphere Research Station on Qinghai-Xizang Plateau, Northwest Institute of Eco-Environment and Resources, Chinese Academy of Sciences, Lanzhou 730000, China
[5]Qinghai Institute of Meteorological Science, Xining 810001, China
[6]32016 PLA Troops, Lanzhou 730000, China
[7]School of Information Science and Engineering, Lanzhou University, Lanzhou 730000, China

*Correspondence*: Lihui Luo (luolh@lzb.ac.cn)

**Abstract.** There exists a narrow engineering corridor across the Qinghai-Tibet Plateau (QTP) with widely distributed slopes, called the Qinghai-Tibet Engineering Corridor (QTEC), where a variety of important infrastructure is concentrated. These facilities are the transportation routes for people, materials, energy, etc. from mainland China to Tibet. From Golmud to Lhasa, the engineering corridor covers 632 kilometers of permafrost containing the densely occurring Qinghai-Tibet Railway, Highway, and power/communication towers. Slope failure in permafrost regions, caused by permafrost degradation, ground ice melting, etc., and affects the engineering construction and permafrost environments in the QTEC. We implement a variety of sensors to monitor the hydrological-thermal deformation between the permafrost slopes and permafrost engineering projects in the corridor, and the aforementioned sensors are densely located on or around the permafrost slopes. In addition to soil temperature and moisture sensors, the global navigation satellite system (GNSS), terrestrial laser scanning (TLS), and unmanned aerial vehicles (UAVs) were adopted to monitor the thermal deformation spatial distribution and changes. An integrated dataset of hydrological-thermal deformation in permafrost engineering and slopes in the QTEC between 2014 and 2019, including meteorological and ground observations, TLS point cloud data, and RGB and thermal infrared (TIR) images, can be of great value for estimating the hydrological-thermal impact and the stability between engineering and slopes under the influence of climate change and engineering disturbance. The dataset and code were uploaded to the Zenodo repository and can be accessed through https://zenodo.org/communities/qtec, including Meteorological and ground observations at http://doi.org/10.5281/zenodo.3764273 (Luo et al., 2020e), TLS measurements at http://doi.org/10.5281/zenodo.3764502 (Luo et al., 2020b), UAV RGB and TIR images at



http://doi.org/10.5281/zenodo.3764280 (Luo et al., 2020c), and R code for permafrost indices and visualization at

http://doi.org/10.5281/zenodo.3766712 (Luo et al., 2020d).

## 1 Introduction

Permafrost is frozen soil or rock containing ice and organic material remains at or below 0 ℃ for at least two consecutive years and occurs mostly in the Northern Hemisphere, Alaska, and Qinghai-Tibet Plateau (QTP) (Wang et al., 2018;Zhang et al., 1999). As a typical mountain permafrost region, permafrost slopes occur widely spread across the QTP. Inevitably, much

of the QTP infrastructure is built on these slopes (Luo et al., 2018c;Jin et al., 2008). There exists a narrow engineering corridor on the QTP, where a variety of important infrastructure is concentrated, called the Qinghai-Tibet Engineering Corridor (QTEC) (Luo et al., 2018b;Zhang et al., 2015). These facilities are the transportation routes for people, materials, energy, etc. from mainland China to Tibet, and the QTEC is several hundred meters to several kilometers wide. From Golmud to Lhasa across the QTP, more than 1120 kilometers of the engineering corridor contains the densely occurring

Qinghai-Tibet Highway (QTH), Golmud-Lhasa pipeline, Lanzhou-Xining-Lhasa fiber optic cable, Qinghai-Tibet Railway (QTR), and 440-kV Qinghai-Tibet DC networking system, which were completed and opened in 1954, 1977, 1997, 2006 and 2013, respectively (Jin et al., 2008). There are more than 6 major linear projects in the QTEC, covering 632 kilometers of permafrost and approximately 550 kilometers of continuous permafrost, with widely distributed slopes (Luo et al., 2018b). Against the background of global climate change and increasing human activities, permafrost degradation is remarkable,

resulting in an increase in the number of permafrost disasters (Weber et al., 2019;Luo et al., 2020a). In engineering around permafrost slopes, permafrost disasters have widely occurred (Ma et al., 2006). Slope instability-type disasters tend to occur frequently, thereby causing direct or potential harm to the engineering projects in this region, which has become the main problem affecting the safe operation and service performance of these engineering projects while increasing the difficulty and cost of engineering maintenance (Niu et al., 2015;Wirz et al., 2015). Engineering in permafrost areas will inevitably

destroy the surface energy and moisture balance, resulting in higher temperatures at the top of the permafrost (TTOP) and a lower permafrost table (Sun et al., 2019;Zhang et al., 2020). Permafrost engineering occurs on slopes, which changes the circulation of surface and suprapermafrost water causing thermal erosion (Han et al., 2019;Mu et al., 2018). Permafrost is affected not only by the construction and operation of engineering but also by the long-term heat impact of climate change (Wicky and Hauck, 2017;Gruber and Haeberli, 2007). In particular, high temperatures and highly ice-rich permafrost with a

mean annual air temperature (MAAT) above -1.0 ℃ and an ice volume content higher than 25% are more sensitive to climate change and disturbances in engineering activities (Wu and Zhang, 2008;Patton et al., 2019). In the past 60 years, the MAAT of the seasonal and island frozen soils along the QTEC has increased 0.3 to 0.5 ℃, and the MAAT in the continuous permafrost area has increased 0.1 to 0.3 ℃ (Obu et al., 2019). The QTP is a large-scale amplifier of global change, experiencing warming above the global average. If the air temperature rises 2.6 ℃, permafrost with a mean annual ground

temperature (MAGT) higher than -1.0 ℃ will degrade to seasonal frozen soil after 50 years (Luo et al., 2018c;Wu et al.,





2002). Furthermore, the permafrost slope in the freezing or/and thawing process is subject to frost heave or/and thawing subsidence, which results in deformation and even destruction of the foundation of engineering facilities, thereby in turn affecting the normal use of these engineering facilities (Luo et al., 2017;Streletskiy et al., 2019). In ice-rich slopes, with melting underground ice due to rising temperatures, the cohesion and angle of internal friction between the active layer and

underground ice decrease and become extremely unstable under the influence of gravity (Yuan et al., 2017). The instability of permafrost slopes will lead to degradation of the permafrost environment, including the melting of underground ice, ground temperature increase, habitat fragmentation, etc., and if these slopes are located near permafrost engineering such as railways and highways, the direct damage of thaw slumps, frost heave, landslides, rockfalls, etc. may cause great harm to the stability of permafrost engineering (Niu et al., 2015;Luo et al., 2018a).

The Chinese National Highway Network Plan, released in 2004, is aimed at building the Beijing-Lassa Expressway, which will be the only expressway to enter Tibet. The Beijing-Tibet Expressway will also be built within this narrow QTEC and will run parallel to or cross-positioned to the already crowded corridor (Ma et al., 2017). The interaction between the dense layout of various permafrost engineering facilities and permafrost slopes cannot be ignored. The stability and potential disaster of permafrost slopes constitute an area of increased interest of the international permafrost community, but studies

on the spatiotemporal dynamic change in the unstable permafrost slopes in the QTEC are still lacking. A thorough understanding of the impact of a variety of permafrost engineering infrastructure (QTH, QTR, power/communication towers, etc.) that are densely distributed on or around permafrost slopes is lacking. Therefore, it is important to evaluate the interaction and influence between permafrost slopes and permafrost engineering by integrating hydrological-thermal deformation monitoring (Luo et al., 2018a).

Terrestrial laser scanning (TLS) with the global navigation satellite system (GNSS) was used to monitor the deformation of permafrost slopes (Luo et al., 2017;Arenson et al., 2016) and steep bedrock permafrost (Weber et al., 2019) to construct a high-resolution digital terrain model of the permafrost region (Boike et al., 2019). Unmanned aerial vehicles (UAVs) can be equipped with different sensors, such as visible digital, thermal infrared (TIR) and multispectral sensors, to estimate the spatial distribution of the ground surface temperature on permafrost slopes and evaluate the thermal influence of nearby

engineering infrastructure (Luo et al., 2018a). UAVs can be used to monitor vegetation and terrain information in permafrost regions (Léger et al., 2019). Sluijs et al. (2018) also adopted UAVs to quantify the deformation dynamics of thaw slumps and to estimate the transfer volumes of sediment.

We provide an integrated dataset of the hydrological-thermal-deformation dynamics between permafrost engineering and slopes in the QTEC from 2014 to 2019. In addition to using soil temperature and moisture sensors to monitor in situ

hydrothermal changes, GNSS, TLS, and UAVs were also adopted to observe the spatial distribution of thermal deformation. This synthesis dataset for permafrost engineering and slopes includes measured air and ground temperatures and moisture at depth intervals up to 2.6 m, MAAT, MAGT, TLS point cloud data, and RGB and TIR images. To fully understand and leverage existing datasets and to allow for full transparency and repeatability, we provide comprehensive information and metadata, including complete documentation of the dataset and technical methods.



## 2 Site description

The study area is located in the Kunlun Mountain Pass (KMP) in the QTEC, where the QTH, QTR, and power/communication towers are crisscrossed (Luo et al., 2018a). They are densely situated across a width of 200 m, and a variety of projects occurs on permafrost slopes connected by supporting bridges or laying towers or directly on roadbeds. This location is excellent to observe the effects of climate change and engineering operations on the hydrothermal deformation of permafrost slopes. The KMP is located in the hinterland of the QTP and in the middle of the East Kunlun Mountains, adjacent to the giant main fault in the northern part of the plateau, which controls its formation and development (Wu et al., 2017). In the KMP, thick early Quaternary sediments were deposited, and the sedimentary strata recorded the history of tectonic evolution in the northern part of the QTP. The MAAT in the KMP is -4 ℃, and the mean annual ground surface temperature (MAGST) ranges from -2.4 ℃ to 3.2 ℃ at a depth of approximately 15 m below the ground surface. The monthly air temperature ranges from a minimum of -28 ℃ in winter to a maximum of 18 ℃ in summer. The mean annual total precipitation ranges from 300–500 mm, and precipitation occurs most frequently from May to September, accounting for 80% of the annual total precipitation, with the highest precipitation occurring from July to August. Permafrost is more developed and has a high ice content with an active layer thickness of more than 3 m and is generally present above 4200 m in the region. Permafrost can be found in the mountains and basins, but most of the valleys do not contain permafrost. Moreover, the permafrost thickness is 92 m, the temperature gradient within the frozen soil is 3.45 ℃/100 m, and the geothermal gradient below the frozen soil is 3.9 ℃/100 m (Luo et al., 2019).

Two slopes (slopes A and B) within the KMP were selected (Figure 1). Slope A (94°03′46″ E, 35°39′4″ N, 4759 m above sea level (a.s.l.) in elevation) is located along the side of the QTH, while the QTR, supported by bridges, runs across slope B (94°03′48″ E, 35°38′45″ N, 4770 m a.s.l. in elevation), and the QTH is next to slope B. A large number of similar permafrost slopes have been found along the QTEC. The surroundings of these two slopes contain several major projects in the corridor, which is an excellent test site for studying the hydrological-thermal deformation interaction and dynamics between engineering infrastructure and slopes.

Through drilling data, the active layer is mainly composed of clay from the top of the soil to a depth of 270 cm, including wet silty clay from 0 to 50 cm, loose silty clay from 50 to 120 cm, compacted clay from 120 to 180 cm, saturated clay mud from 180 to 230 cm, loose silty clay with gravel from 230 to 250 cm and thick underground ice from 250 to 270 cm. According to the soil profile, the types of permafrost on the above two slopes can be classified as ice-rich permafrost, and the ATL here is generally larger than 3 m. The vegetation type in this area is a typical alpine desert steppe. The vegetation on slope A is sparse, at a coverage lower than 5%, and most of the vegetation is gathered at the top of the slope, while slope B contains almost no vegetation.

From 2014-2017, four deformation monitoring campaigns using TLS with GNSS and soil hydrothermal in situ monitoring were deployed. Moreover, a UAV system equipped with RGB and TIR sensors was adopted to monitor the spatiotemporal changes in the ecological environment and ground surface temperature of the slopes. The deployment of these instruments




provides an integrated dataset for hydrological-thermal deformation monitoring of permafrost slopes and the various frozen soil projects (QTR, QTH, and power/communication towers) located on (middle) or near them.



**Figure 1**. Geography of the study area, including the two permafrost slopes A and B, Qinghai-Tibet Highway and Railway, and power/communication towers.

## 3 Data description

### 3.1 Meteorological observations

Observation of meteorological factors was conducted at two permanent meteorological stations (Golmud and Wudaoliang) and one field meteorological station (Xidatan) with daily meteorological records for the period of 2014-2018 (Table 1), which were retrieved from the National Meteorological Information Center (NMIC, http://data.cma.cn). The two weather stations are located on the north and south sides of the two slopes and are also the closest national weather stations to the two slopes. The meteorological database includes the daily mean, maximum (max) and minimum (min) air temperatures, wind

speed, observed and corrected precipitation, evaporation, air humidity, atmospheric pressure, sunshine duration, daily mean, max, and min ground surface temperatures, and soil temperature at different depths (i.e., 5, 10, 15, 20, 40, 50, 80, 160, and 320 cm). Golmud and Wudaoliang are two national meteorological stations in China, while Xidatan is a national field station. Although these two national meteorological stations are far from the above two slopes, the data obtained from these meteorological stations are very valuable in this harsh environment. Their data can be combined with the data obtained from

the Xidatan field station to analyze the spatiotemporal dynamics of the permafrost slopes in the corridor.

Meteorological station Golmud (36°25′ E, 94°54′ N; 2808 m a.s.l.) is located in the urban area of Golmud, south of the two slopes, with few surrounding buildings. This weather station area is located in the seasonal frozen soil zone. The distance between the station and the two slopes is approximately 115 km.

Meteorological station Xidatan (35°43′ E, 94°08′ N; 4538 m a.s.l) is located in the northern part of the two slopes and is the

closest field weather station to the two slopes. The distance between the station and the two slopes is approximately 9.7 km. The Xidatan field station is 300 m away from the QTH and is located at the northernmost end of the permafrost in the QTEC. The vegetation type on the underlying surface is an alpine meadow. Figure 2 shows the data for this station. Observation data was retrieved from the National Tibetan Plateau Data Center (NTPDC, https://data.tpdc.ac.cn/).

Meteorological station Wudaoliang (35°13′ E, 93°05′ N; 4612 m a.s.l.) is located to the north of the two slopes in a

continuous permafrost zone, next to the 109 National Highway along the Qinghai-Tibet Highway. The distance between the station and the two slopes is approximately 101 km.

**Table 1**. List of the meteorological observation data, where n/a indicates not applicable. Automated observations were conducted at the Golmud and Wudaoliang stations in 2000 and 2001, respectively.

| SID | Station name | Latitude | Longitude | Elevation | Automatic station | From year | To year | Source |
| --- | --- | --- | --- | --- | --- | --- | --- | --- |



| | | | | | model | | | |
|---|---|---|---|---|---|---|---|---|
| 52818 | Golmud | 94°54′ | 36°25′ | 2808 | Vaisala Milos 500 | 1955 | 2018 | NMIC |
| XTDMS | Xidatan | 94°08′ | 35°43′ | 4538 | n/a | 2014 | 2018 | NTPDC |
| 52908 | Wudaoliang | 93°05′ | 35°13′ | 4612 | Vaisala Milos 500 | 1956 | 2018 | NMIC |


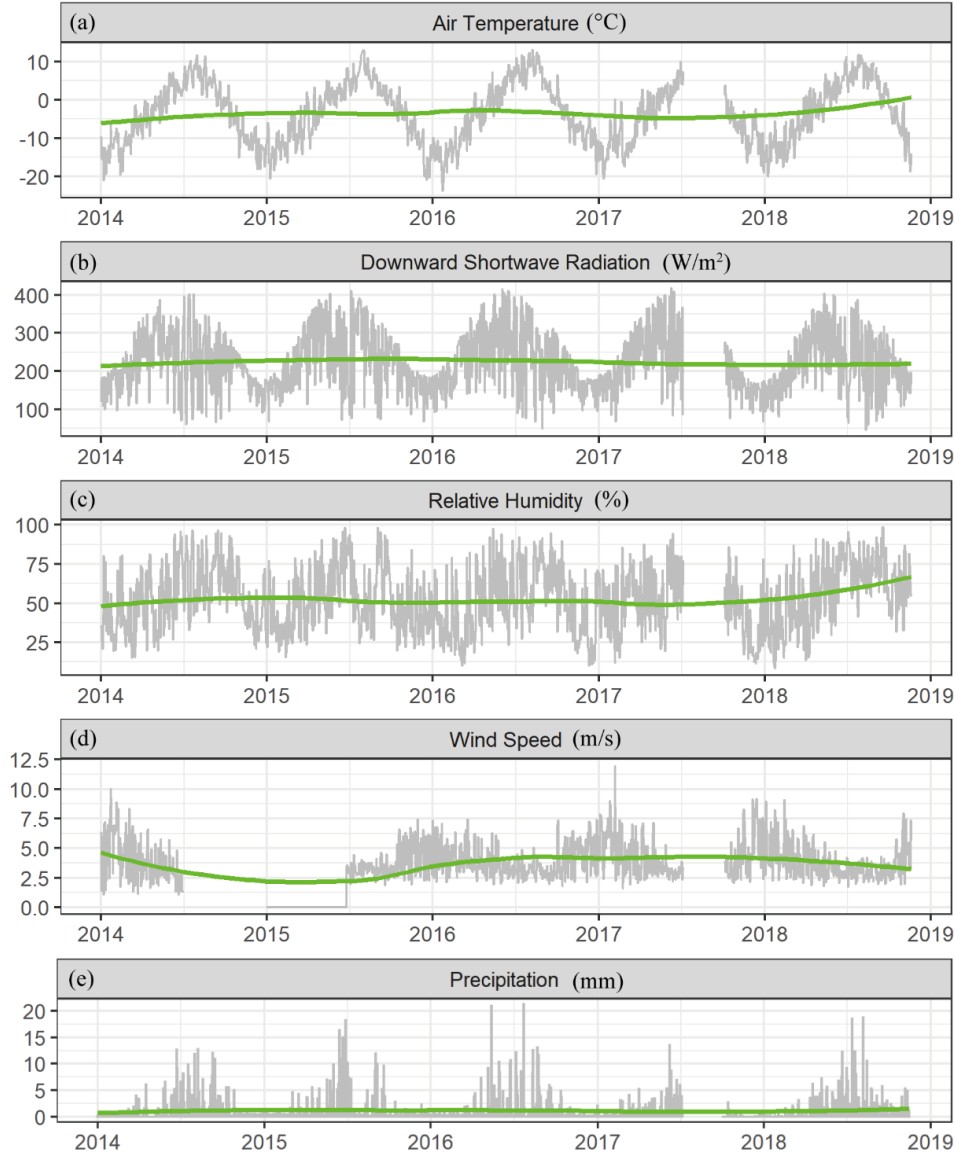

**Figure 2**. Time series (daily mean values) of the Xidatan field station from 2014-2018: (a-e) meteorological observation data.

### 3.2 Ground observations

Changes in soil temperature and humidity can be used to indicate the water and heat transfer processes, which will strongly affect the physical and mechanical properties of frozen soil, which will further affect the stability of slopes. The ground temperature and moisture data from the near-surface to within 270 cm in the active layer were recorded. In situ ground observations were deployed starting in July 2013 in the active layer of slope A at 11 depths (1, 30, 63, 80, 100, 123, 140, 175, 205, 235, and 260 cm) using thermocouple probes (105T, Campbell Scientific) to measure the soil temperature and using 11

time-domain reflectometer (TDR) probes (model CS615-L, Campbell Scientific) at 11 depths (10, 20, 48, 74, 91, 110, 135, 157, 190, 220, 245, and 270 cm) to measure the soil volumetric water content from 2014-2018 (Figure 3). The TDR probes were mounted horizontally along the soil profile next to the temperature probes at the different soil depths, and measurements were recorded once per hour. Since the soil undergoes a freezing period from refilling to compaction, the 2013 data are not analyzed. A Campbell Scientific CR1000 data logger is used to connect the ground temperature and

volumetric water content probes. Figure 4 shows the Kriging interpolation data of the soil temperature and volumetric water content at the different depths from 2014-2019.

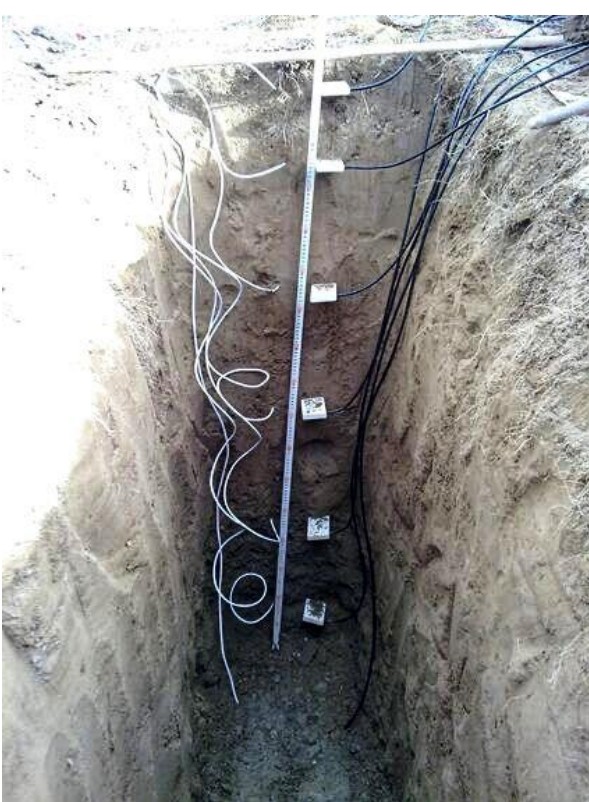

**Figure 3**. Ground sensor installation.

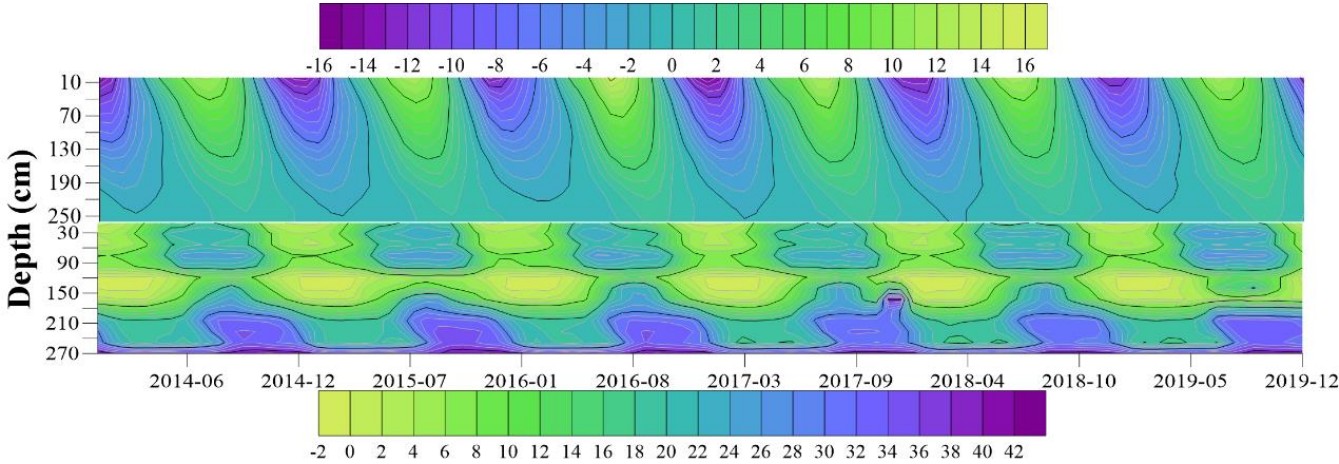


**Figure 4**. Soil temperature and volumetric water content from 2014-2019. (a) Soil temperature (℃); and (b) soil moisture (%).

### 3.3 TLS and GNSS

Deformation monitoring was performed through TLS with network real-time kinematic (RTK) service provided by the National Geodetic Control Network (NGCN) for the China Geodetic Coordinate System 2000 (CGCS 2000) at permanent reference stations for the GNSS (Figure 5a and 5b). A FARO Focus³ᴰ X130 3D laser scanner and six Trimble 5700 GNSS systems were deployed at the study site between May 2014 and October 2015. Two GNSSs were adopted as datum points 30 km outside the study area (Figure 5c and 5d), and another four GNSSs were deployed as reference points around one of the

slopes. According to the manufacturer specifications, the FARO TLS instrument measures 3D coordinates with a distance accuracy up to ±1 mm, and the ranging error is ±2 mm. The Trimble 5700 GNSS systems achieve a measurement accuracy of ±5 mm + 0.5 ppm root mean square (RMS) horizontally and ±5 mm + 1 ppm RMS vertically for static and FastStatic GPS surveying, respectively.

Since May and October are the transition periods of the freeze-thaw cycle, we conducted monitoring campaigns in the

months of May and October between May 2014 and October 2015. The successive three freeze-thaw phases are referred to as the first thaw (May 2, 2014, to October 10, 2014), first freeze (October 10, 2014, to May 3, 2015) and second thaw (May 3, 2015, to October 4, 2015). The above two slopes mainly manifested melting collapse during thawing and frost heave during freezing, but after several freeze-thaw cycles, frost heave dominated (Figures 6 and 7). A full slope scan requires approximately 30 scan positions to generate 3D point cloud data. Each scan requires the placement of six white reference

spheres at the study site, and they are visible at all scan locations. Two to three of these reference spheres are moved for the next scan. Six reference sphere data points can be combined with the GNSS data to georeference, register, align, and mosaic the point cloud data of the TLS instrument with FARO SCENE and Geomagic Studio software.

As a supplement to the TLS point cloud data, we prepared Sentinel-1 deformation data for the freeze-thaw stage in the study area from 2014 to 2020 using interferometric synthetic aperture radar (InSAR) technology.

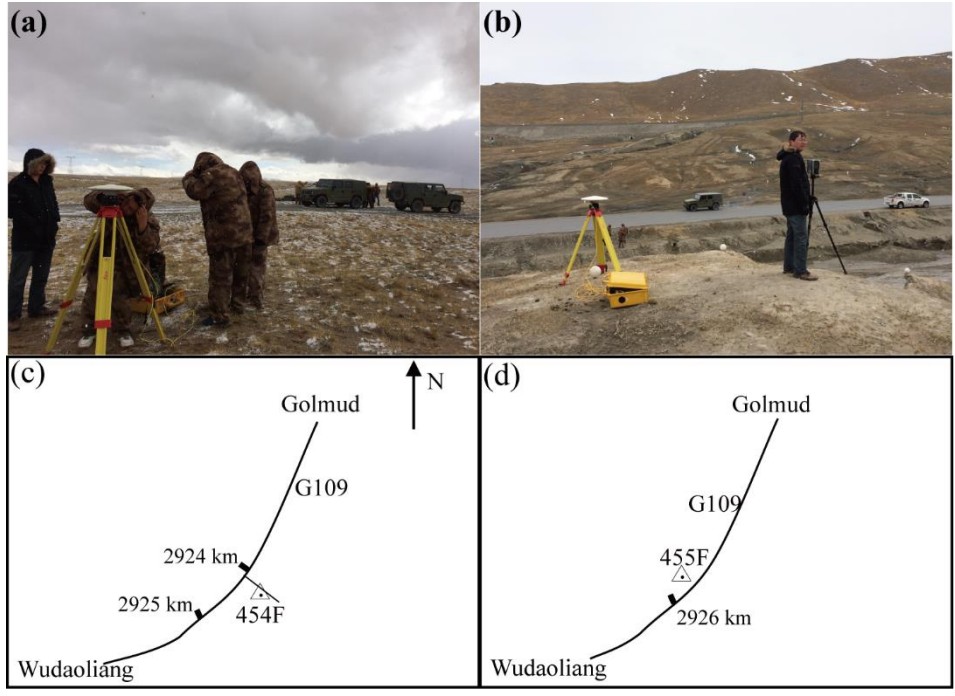


**Figure 5.** TLS observation with the GNSS. (a) Datum station of the GNSS; (b) TLS observation with the GNSS and white reference sphere set; (c) datum station 454F; (d) datum station 455F.

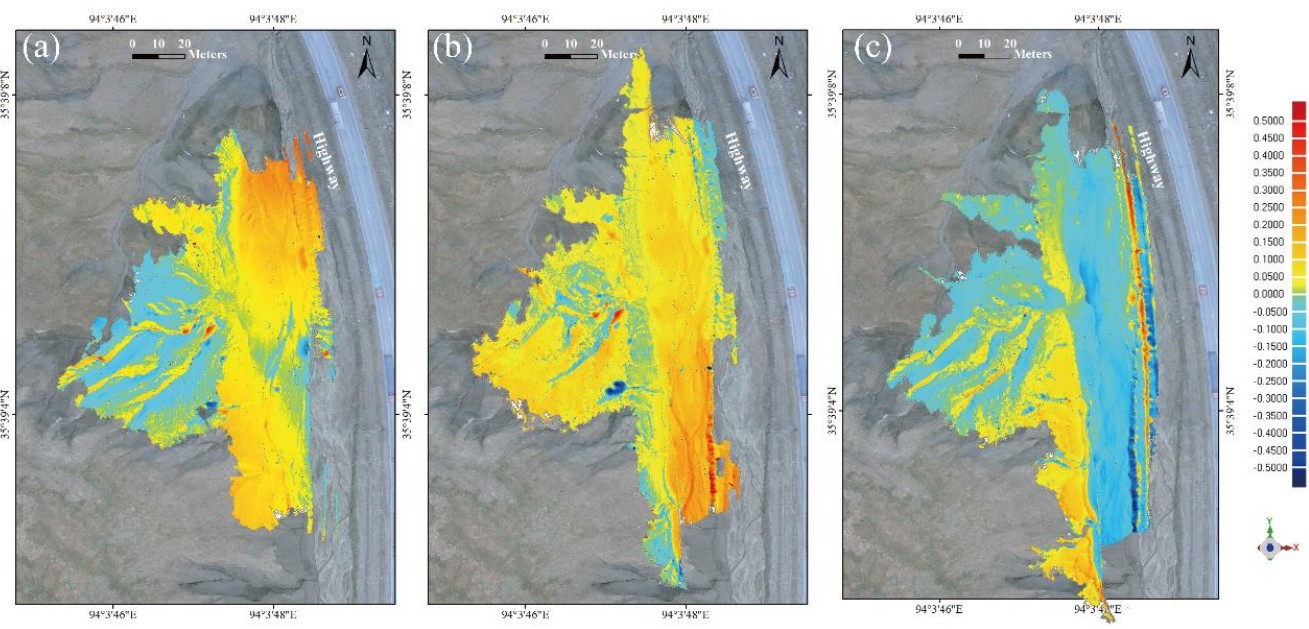



**Figure 6.** Deformation of slope A during the first thaw, first freeze, and second thaw. (a) First thaw; (b) first freeze; (c) second thaw.

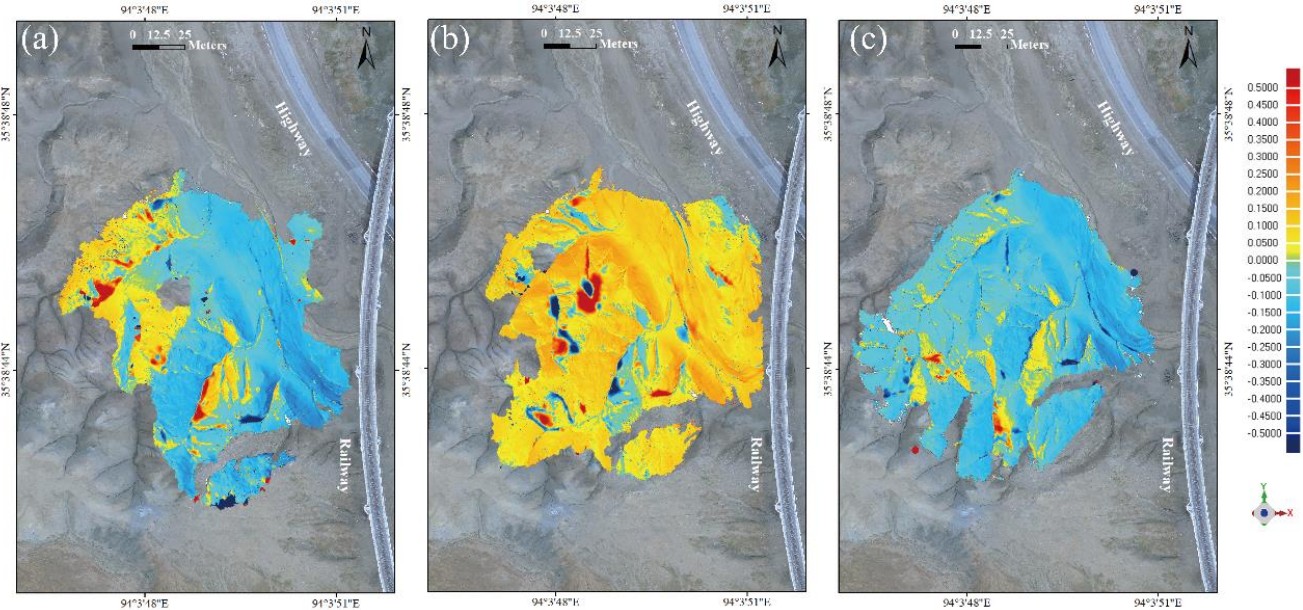

**Figure 7.** Deformation of slope B during the first thaw, first freeze, and second thaw. (a) First thaw; (b) first freeze; (c) second thaw.

### 3.4 UAV with multisensors

The change in frozen soil is greatly affected by the temperature. To monitor the heat exchange between the two slopes and the engineering infrastructure around them, a UAV system with mounted RGB and TIR sensors was adopted (Luo et al., 2018a). The DJI Inspire 1 UAV system (DJI, Inc., Shenzhen, China) weighs approximately 2.85 kg including propellers and batteries and is equipped with a Zenmuse X3 RGB camera and a Zenmuse XT TIR sensor. The camera and sensor weigh 221 g and 270 g, respectively. The UAV is also equipped with GPS and an inertial measurement unit (IMU) to measure the geographical and flight positions, respectively. The TIR sensor has a resolution of $336 \times 256$ pixels, a thermal sensitivity of < 0.05 ℃ at f/1.0, a field of view of 17 ° (H) $\times$ 13 ° (V), a focal length of 19 mm, and a spectral response in the electromagnetic spectrum. The range of the above TIR sensor is 7.5–13.5 μm. A mobile phone, equipped with a flight control system app, is used to control the flight of the UAV, in addition to DJI GO and Pix4Dcapture software. Regardless of the RGB or TIR sensor, the JPG data format is selected. Both instruments adopt vertical angles to capture images, with RGB overlap rates above 75% and TIR overlap rates above 80%.

Figure 8 shows the drone flight over the slopes. For these two slopes, we conducted two flight experiments with UAVs and onboard RGB sensors in 2016 and 2017. The two experimental datasets were processed with pix4dmapper software to generate digital elevation models (DEMs) and orthorectification images. In 2016 and 2017, two flight experiments with



UAV-mounted TIR sensors were performed. Regarding the TIR flight experiments, each experiment lasted from morning to afternoon, with intervals of 1 to 2 hours (Table 2). The TIR data formats selected for 2016 and 2017 were TIF and JPG, respectively. The TIR images were processed with software program FLIR Tools (FLIR System Inc., USA) (Figure 9), and the ground surface temperature differences were analyzed to determine the effects of the different permafrost engineering

240 operations on the slopes (Figure 10). This study analyzes the thermal impact of engineering operations on permafrost slopes. The results show that the QTH has the greatest thermal impact on permafrost slopes, followed by the QTR and finally the power/communication towers.

**Table 2.** UAV observations during the 2016–2017 study period.

| Flight Date | Flight Time | Height | Slope | Sensor |
|---|---|---|---|---|
| yyyymmdd | hh:mm | m | | |
| 20160417 | 13:36-13:56 | 20-120 | Slopes A and B | RGB |
| 20160830 | 10:18-13:55 | 120 | Slopes A and B | RGB |
| 20170822 | 11:26-13:46 | 120 | Slopes A and B | RGB |
| 20160830 | 12:47-12:52 | 30 | Slope A | TIR |
| 20170722 | 11:00-15:51 | 150 | Slopes A and B | TIR |
| 20170823 | 10:30-17:25 | 150 | Slopes A and B | TIR |

245

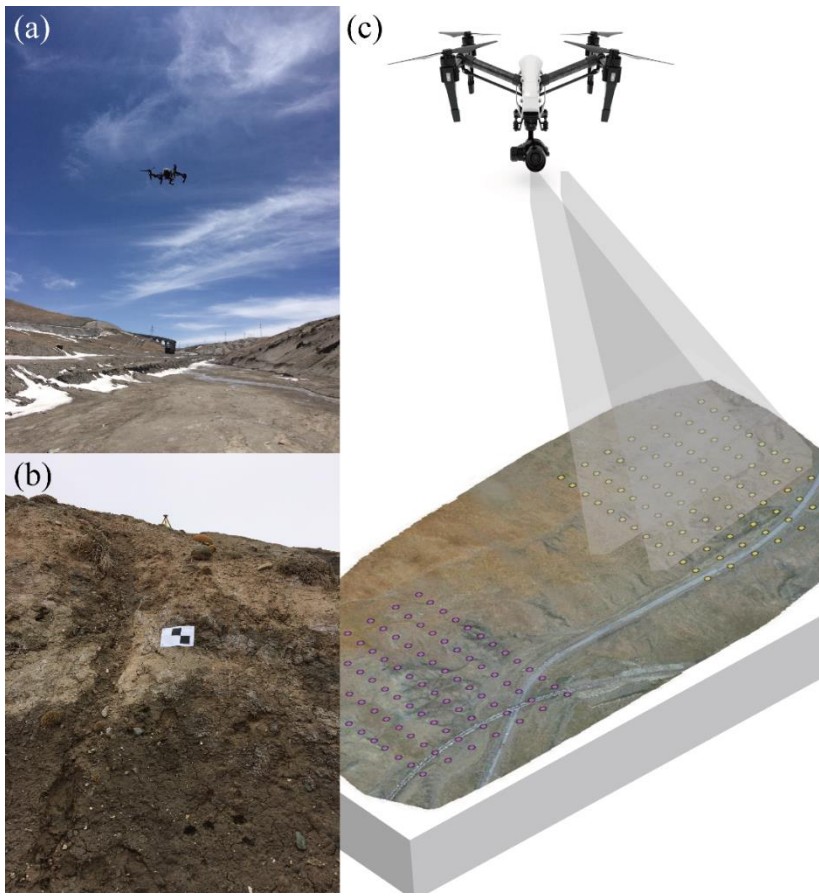

**Figure 8**. UAV observations. (a) UAV observation; (b) GNSS and rectangular white–black cardboards; (c) UAV flight path over the two permafrost slopes.



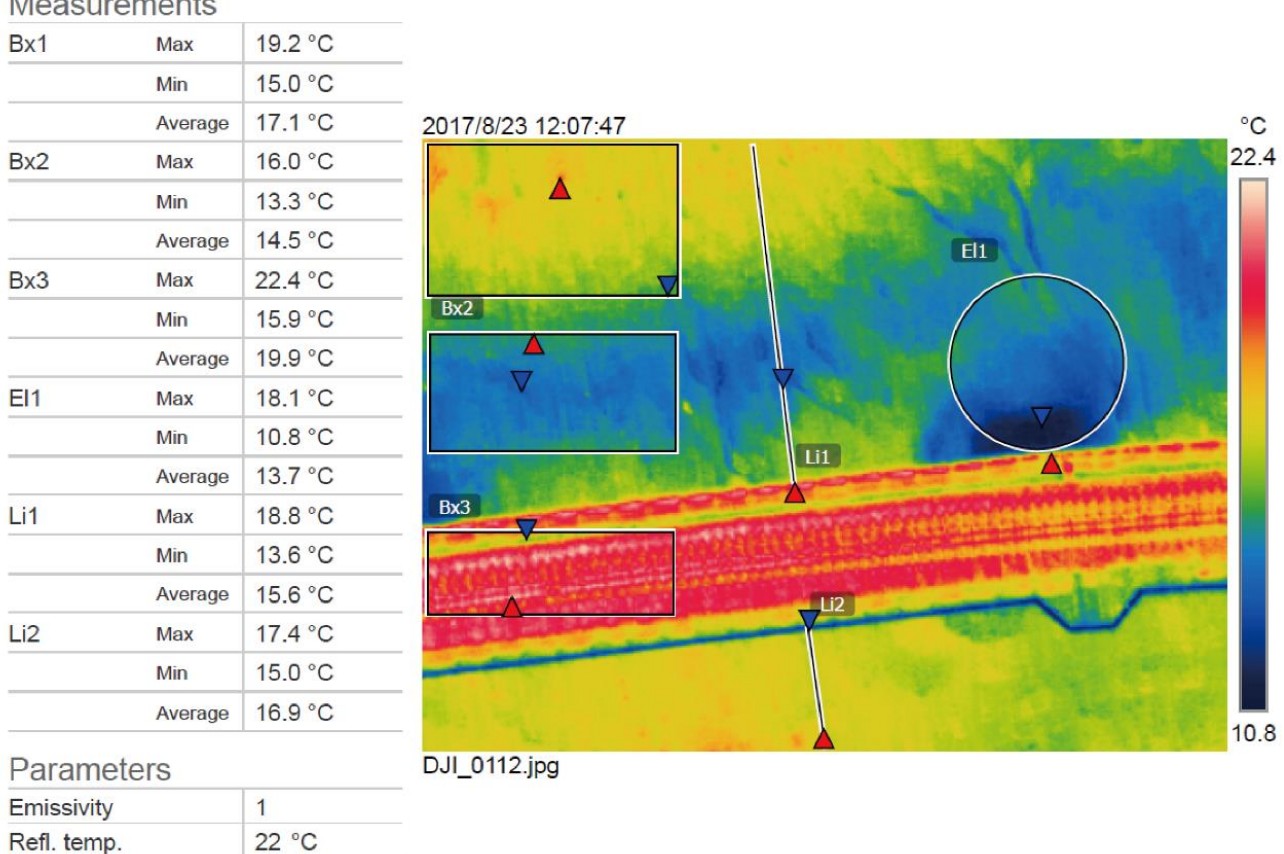

250 **Figure 9**. Analysis of the ground surface temperature using the UAVs with TIR sensors and the Qinghai-Tibet Railway as an example.

**Figure 10**. Ground surface temperature using the UAVs with TIR sensors. (a); Qinghai-Tibet Highway; (b) power/communication tower; (c) slope A; (d) slope B.

## 4 Data quality control

The meteorological data have been quality controlled, and all suspicious and incorrect data have been manually re-examined and corrected. Ultimately, all feature data are marked with a quality control code (Table 3). In terms of the instantaneous meteorological values, if data are missing due to collector or communication issues, the terminal directly generates missing data when the terminal commands the data input, and the corresponding quality control identifier is 8. If there are no missing data, the quality control is assessed to be incorrect at the terminal. When the command data are output, values are still generated, and the corresponding quality control identifier is 2, but the erroneous data does not participate in the subsequent related calculations or statistics. After quality control, the availability of the various weather elements is usually higher than 99%, and the correct data transmission rate approaches 100%.





The use of the GNSS and white reference spheres can improve the accuracy of TLS monitoring, while the use of the GNSS
and rectangular white–black cardboards can improve the accuracy of UAV monitoring (Luo et al., 2019). By measuring the
ground reference and control points of the TLS, the GNSS is used to ensure the orientation and registration of the different
3D datasets in the common coordinate system. The spherical shape achieves the maximum scanning efficiency in all
directions and has proven to be the most effective laser scanning target, which can be used in conjunction with the GNSS
datum points (454F and 455F) and control points to register, align and mosaic the TLS data. These GNSS instruments collect
data on the ground reference points to ensure the maximum geospatial accuracy, and they are subject to stringent ground
controls to reference and calibrate the 3D FARO laser scanner. Moreover, the 3D laser scanner and GNSS obtain continuous,
high-precision spatial deformation data on the slopes, and we can compare the spatial changes over time through the GNSS
network. These targets are used for registration and as georeferences and checkpoints. Therefore, their positioning accuracy
directly affects the accuracy of the data processing results. Data preprocessing is proposed to determine the scope of the
slope, filter any noise points and repair data gaps. Semiautomatic and manual processing is conducted to filter the noise
points and repair the gaps in the point cloud datasets (Luo et al., 2019). Due to the high moisture content in the lower part of
the slope, monitoring is easily disturbed by vibrations, resulting in noise. At the top of slope A, large wild animals such as
wild donkeys were observed. Therefore, when analyzing its deformation, these aspects need to be considered.

**Table 3**. Quality control codes for the meteorological station data. The variable names are suffixed with _QC.

| Quality control code | Description |
| --- | --- |
| 0 | Correct data, no modification |
| 1 | Suspicious data, no modification |
| 2 | Error data, no modification |
| 3 | Missing data, no modification |
| 4 | Data with revised values |
| 5 | Originally suspicious data, has been modified |
| 6 | Originally error data, has been modified |
| 8 | Originally missing data, has been modified |
| 9 | No data quality control |

**5 Data and code availability**

All data and R code presented in this paper are available from Zenodo (https://zenodo.org/communities/qtec; last access for
all Zenodo links: 29 April 2019), which provides a dataset view and download statistics. The data and code with open access
(including links to the subsets) can be found on either repository via the following links:



• Meteorological and ground observations:

http://doi.org/10.5281/zenodo.3764273 (Luo et al., 2020e)

• TLS measurements:

http://doi.org/10.5281/zenodo.3764502 (Luo et al., 2020b)

• UAV RGB and TIR images:

http://doi.org/10.5281/zenodo.1254848 (Luo et al., 2020c)

• R code of permafrost indices and visualization:

http://doi.org/10.5281/zenodo.3766712 (Luo et al., 2020d)

## 6 Summary and outlook

Slope failure in permafrost regions, caused by permafrost degradation and ground ice melting, affects the engineering infrastructure and permafrost environment in the Qinghai-Tibetan Engineering Corridor. Most of the current studies are based on the interaction between individual engineering projects and permafrost slopes by means of multipoint monitoring, interpolation or simulation, but the dense layout of the various projects and the fragile and sensitive permafrost slopes in the corridor has rarely been studied before as a whole. The permafrost slopes and various projects (QTR, QTH, and

power/communication towers) located in the corridor are chosen as the research objects, and integrated drone-based visible and TIR remote sensing and in situ monitoring technology are deployed, combined with image mosaic, three-dimensional modeling and spatial analysis. The dataset presented here, including the ground hydrothermal state, spatial ground surface temperature, slope deformation, and meteorological data of the engineering infrastructure and slopes, will be of great value to examine the spatiotemporal hydrothermal dynamics of permafrost slopes under the influence of climate change and

engineering disturbances and to reveal the mutual feedback between the slopes and engineering infrastructure, evaluate the potential hazards of long-term stability and safety operation of the engineering infrastructure and slopes, and provide data support for the safety range and layout of proposed permafrost engineering infrastructure. This dataset contains both site and space features, on both the surface and underground horizons, thus establishing a comprehensive monitoring dataset for the QTP permafrost slopes and surroundings (QTR, QTH, etc.).






## Appendix A: Symbols and Abbreviations

| | |
|---|---|
| CGCS | China Geodetic Coordinate System |
| DSM | Digital surface model |
| GNSS | Global navigation satellite system |
| InSAR | Interferometric Synthetic Aperture Radar |
| KMP | Kunlun Mountain Pass |
| MAAT | Mean annual air temperature |
| MAGST | Mean annual ground surface temperature |
| MAGT | Mean annual ground temperature |
| NGCN | National Geodetic Control Network |
| NMIC | National Meteorological Information Center |
| QTEC | Qinghai-Tibet Engineering Corridor |
| QTH | Qinghai-Tibet Highway |
| QTP | Qinghai-Tibet Plateau |
| QTR | Qinghai-Tibet Railway |
| RTK | Real-time kinematic |
| TDR | Time-domain reflectometry |
| TIR | Thermal infrared |
| TTOP | Temperature at the top of the permafrost |
| TLS | Terrestrial laser scanning |
| UAV | Unmanned aerial vehicle |









**Appendix B: Pictures of the Field Site and Selected Instrument Details**

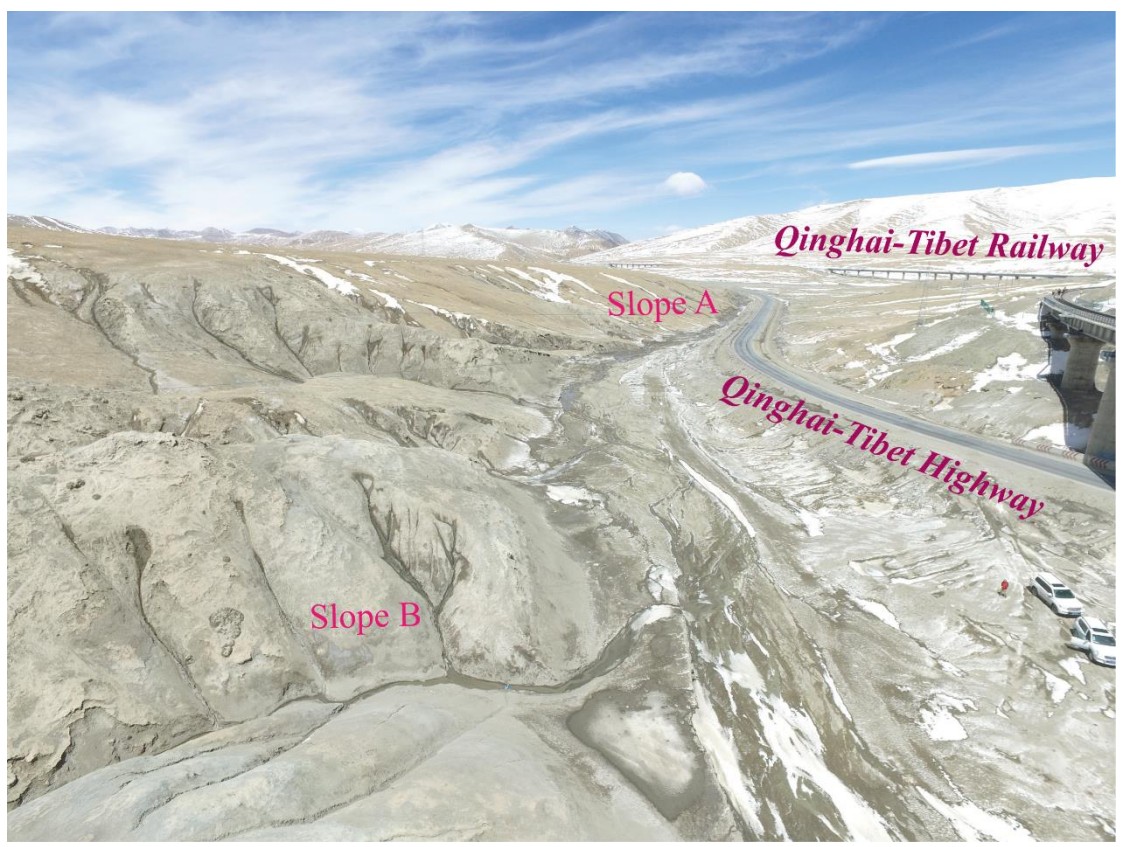


**Figure B1**. Slopes A and B, with the QTH and QTR clearly visible.

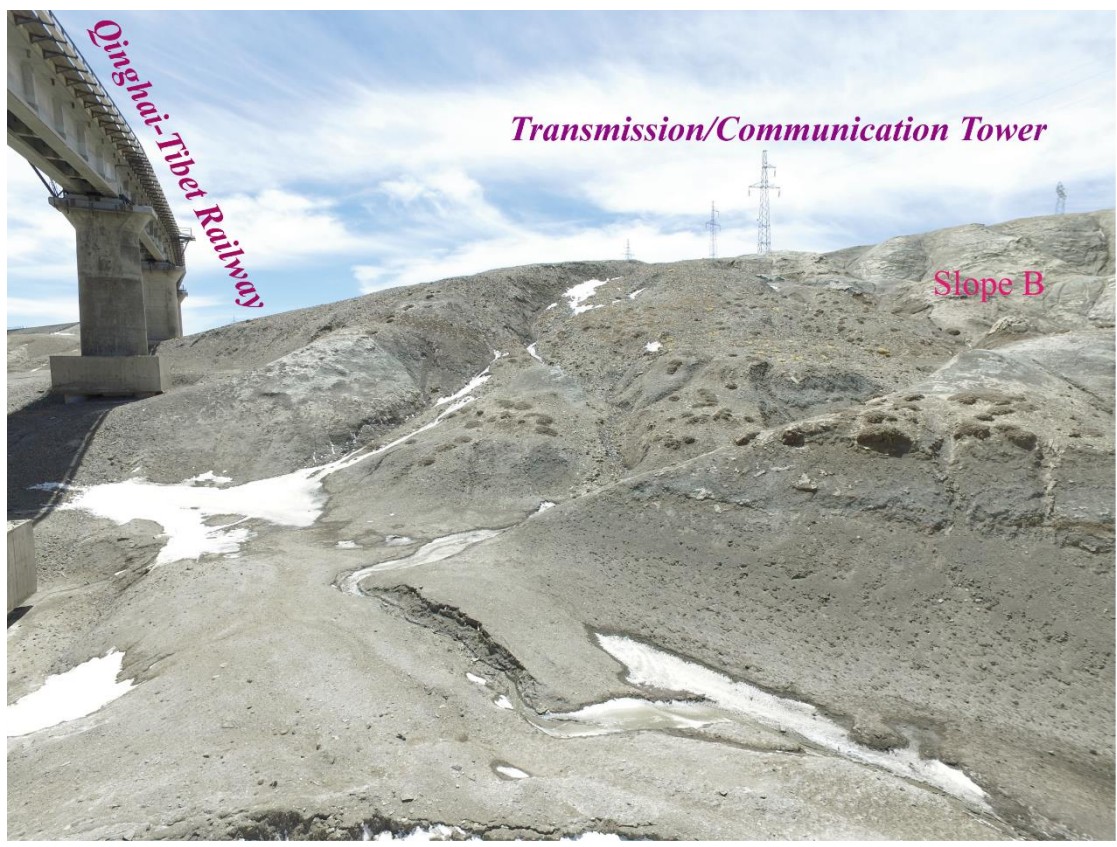

**Figure B2**. Slope B. The railways and power/communication towers are clearly visible.



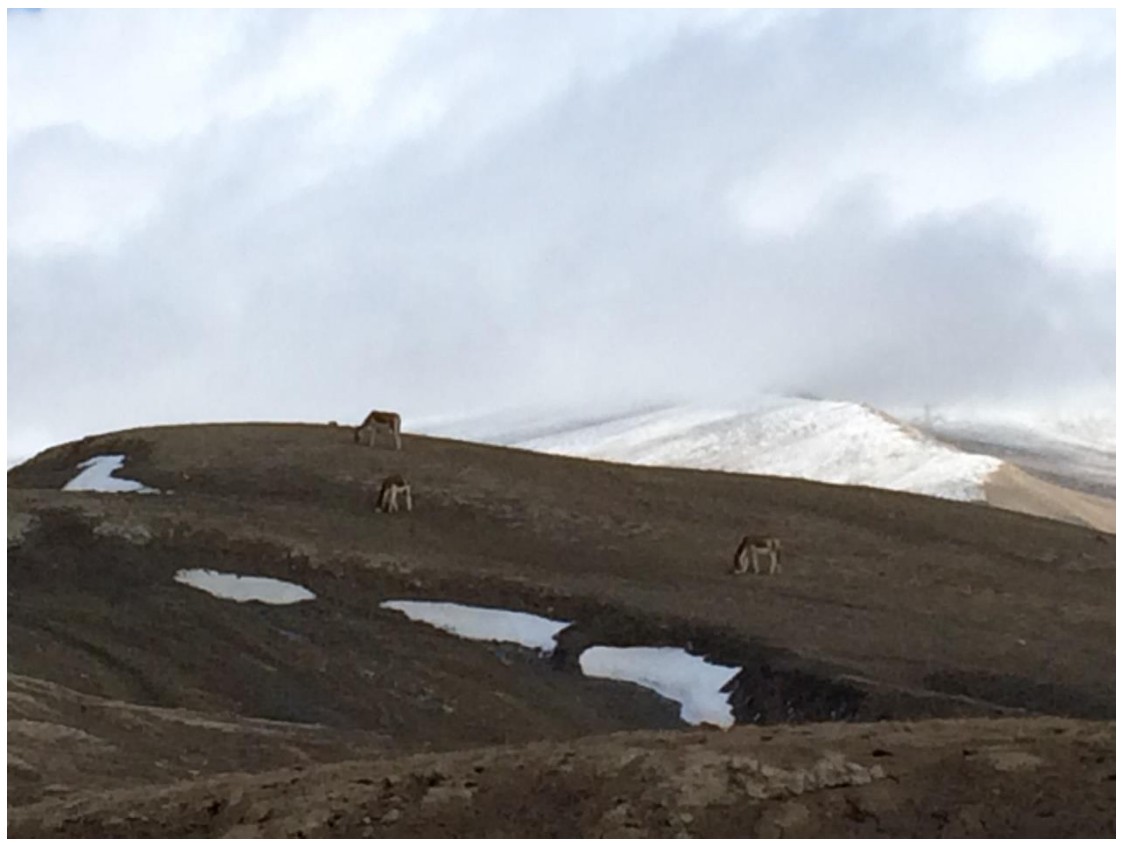

**Figure B3**. There are large animals on the top of slope A.

**Appendix C: Digital Surface Model of the Study Area Using RGB Images**

**Figure C1**. DSM by the UAV. (a) Slope A; (b) slope B.




**Appendix D: Names of the Variables and Units for the Data Files**

**Table D1.** Overview of the meteorological and ground observation data. The suffixes _MIN, _MAX, _AVG, and _QC indicate the minimum, maximum and average values and quality control code of the variable, respectively, while 32766, NA and NAN indicate null values. The suffix of GT with a number indicates the ground temperature in centimeters.

| Variable Name | Description | Unit | Scale |
|---|---|---|---|
| Temperature | Air temperature | ℃ | 0.1 |
| Tmax | Maximum air temperature | ℃ | 0.1 |
| Tmin | Minimum air temperature | ℃ | 0.1 |
| Wind | Wind speed | m/s | 0.1 |
| Precip | Precipitation | mm | 0.1 |
| Corrected_P | Corrected precipitation | mm | 0.1 |
| Evaporation | Evaporation | mm | - |
| Humidity | Air humidity | % | - |
| Press | Atmospheric pressure | hPa | 0.1 |
| Sunshine | Sunshine duration | h | 0.1 |
| GT | Ground temperature | ℃ | 0.1 |




**Appendix E: Physical Parameters of Engineering Infrastructure**

Table E1. Physical Parameters of the engineering infrastructure near the permafrost slopes. There are two types of foundations applied in the construction of the power/communication towers, i.e., * the cone-cylinder spread footing and # the drilled shaft.

| Engineering | Variable | Value |
|---|---|---|
| Highway | Subgrade height | 2.0-2.5 m |
| | Road width | 9 -10 m |
| Railway | Track width | 6.5 m |
| | Bridge diameter | 4.5 m |
| | Bridge height | -16.5 m |
| Tower | Buried base depth | 3.7-5.8 m |
| | Base width of foundation* | 3.5-6.4 m |
| | Diameter of piles# | ~ 1.0 m |
| | Length of piles | 7.0 -16.0 m |

**Appendix F: Permafrost Indices From 1956-2018**

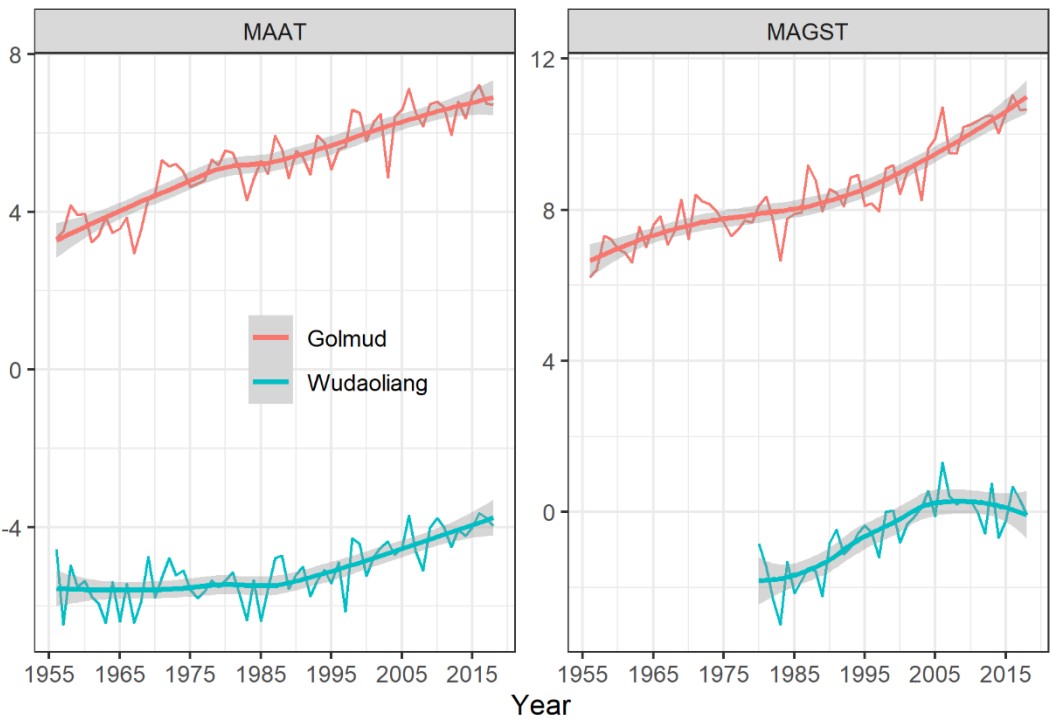

**Figure F1**. Mean annual air temperature (MAAT, ℃) and mean annual ground surface temperature (MAGST, ℃) at meteorological stations Golmud and Wudaoliang.



**Author contributions.** Lihui Luo, Yanli Zhuang, Yanmei Shi and Li Wang initiated and set up the ground observational, TLS, GNSS, and UAV field experiments. Lihui Luo, Yanli Zhuang and Zhongqiong Zhang compiled the database,
performed the analysis, generated the figures and wrote the paper. All authors contributed to the database compilation and assisted in the writing of the paper.

**Competing interests.** The authors declare that they have no conflicts of interest.

**Acknowledgments.** This research was jointly supported by the National Natural Science Foundation of China (41871065), the National Science Fund for Distinguished Young Scholars (41825015), the Key Research Project of Frontier Science of Chinese Academy of Sciences (QYZDJ-SSW-DQC040), and the Strategic Priority Research Program of the Chinese Academy of Sciences (XDA19090122).

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
