# Peer review of "An integrated observation dataset of the hydrological-thermal deformation in permafrost slopes and engineering infrastructure in the Qinghai-Tibet Engineering Corridor"

_Earth System Science Data, 2020_

## Referee Comment (RC1) · Jan Beutel (Referee) · 13 Aug 2020

Dear authors,

This paper gives an overview of measurement data derived from permafrost study sites in the Kunlun Mountain Pass area of the Qinghai-Tibet Plateau, China. The paper describes the locality with focus on the collocated engineered structures of the Qinghai-Tibet highway, railway and power lines. The paper is a companion to data and

processing code published on zenodo.org (Meteo/ground measurements, TLS, UAV images). This paper supersedes further publications by the authors that are based in part of this data.

It is highly appreciated that the authors take the extra effort to collate and describe multiple datasets into one common format and data publication. However, in the present form, the paper is incomplete w.r.t. to a number of details, the metadata describing the data as well as the processing code provided. Two datasets mentioned (Xidatan weather, ground observations, sentinel InSAR data) are not provided. Apart from textual issues I will elaborate below and in the attached commented manuscript pdf file the main issue is that I was not able to run the code in conjunction with the datasets provided. Furthermore some references are missing/misleading. Some of the figures in this paper have already appeared elsewhere (other papers by the authors). Therefore they should be clearly marked as references.

Specific comments:

You are using the term "hydrological-thermal deformation dynamics" and "hydrological-thermal deformation" interchangeably. I understand the first term with dynamics, but am not sure the second is correct. What exactly is a hydrological deformation? I understand thermal deformation (contracting/expansion of a material under thermal stress) and I think I know what you want to say. I would rather talk about permafrost or ground dynamics or ground deformation in the context of landslides or precursory patterns rather than combining process origin (thermal/hydrological) with the observed effect (deformation) in one long term.

Much of your intro argumentation centers around the impact of engineering structures (man made interference through the immediate built environment) on permafrost in QTP and resulting hazards. While this is clearly an important issue a number of references given do not relate to this or should be explained in a different context (see annotations). Also in your data description it is not clear what data are influenced by

engineering (and possibly how much) and what data are not influenced by QTH/QTR etc.

The data should be described concisely with correct metadata. Your data packages and references to the data in the paper do not match, file/directory naming is not explanatory. Please provide a global inventory of the data provided and exact file descriptors. Also it seems your dataset covers data from 1955-2020 (in some parts) but your paper mentions the period 2014-2019. Please clarify. Most importantly the files for ground observations are missing!

You are using a time-domain reflectometer (TDR) probe (model CS615-L, Campbell Scientific) for assessing the soil volumetric content. The probe is specified by the vendor (followup product CS616) for operation in 0-70C only. However you present data in figure 4 down to -16C. https://www.campbellsci.com/cs616-reflectometer Operating Temperature Range $0°$ to $+70°C$

Furthermore • Or, Dani, and Jon M Wraith. 1999. "Temperature Effects on Soil Bulk Dielectric Permittivity Measured by Time Domain Reflectometry: A Physical Model." Water Resources Research 35 (2): 371–83. https://doi.org/10.1029/1998WR900008. • Overduin, Pier & Yoshikawa, Kenji & Kane, D. & Harden, J.. (2005). Comparing electronic probes for volumetric water content of low-density feathermoss. Sensor Review. 25. 215-221. 10.1108/02602280510606507.

detail that it is not at all straightforward to measure these quantities in the frozen state. Therefore I suggest to (1) remove moisture data below T=0C or (2) at least mention that this data must be treated with utmost care as it is outside the spec of the instrument you are using.

Figure 4 should be labeled correctly.

You mention: "This study analyzes the thermal impact of engineering operations 240 on permafrost slopes. The results show that the QTH has the greatest thermal impact on

permafrost slopes, followed by the QTR and finally the power/communication towers."
I can see one figure. But where is the analysis, how is it performed and what is the
quantitative outcome?

The R code provided cannot be used. Please provide comments/readme and explain
the filenames used/origin of the input files. E.g. this file referenced in the code is not
available: xdt <- read.csv("PLOT/XDTMS2014-2018_PLOT.csv", header = TRUE)

Please also note the supplement to this comment:
https://essd.copernicus.org/preprints/essd-2020-106/essd-2020-106-RC1-
supplement.pdf

―――――――――――――――――

---

## Referee Comment (RC2) · Anonymous Referee #2 · 23 Feb 2021

The manuscript by Luo et al. described multiple observation data sets in the Qinghai-Tibet Engineering Corridor (QTEC). I agree with the previous reviewer's comments about the hard-won data in this manuscript. What is particularly commendable is that the author chose a study area where railway, highway and electrical towers are all distributed on a frozen soil slope. Temperature, air and ground temperature, is the most important indicator of changes in frozen soil. The author uses drones equipped with thermal infrared sensors to monitor spatial changes in surface ground tempera-

ture. This data should be relatively rare. This set of data is of great significance for studying the interaction between frozen soil engineering and slopes. Overall, this is a well-prepared manuscript with useful data. The study area is very typical and distinctive. Therefore, I don't have any major suggestions on how to improve the manuscript. Please see some minor comments below.

Minor comments: 1. Please provide a more detailed metadata description of the data set.

2. It is recommended to add the running notes in the code, and increase the readability of the code, so that users can not only execute, but also modify and improve.

3. Please delete Figure B3. If possible, just describe it in the text.

4. The latest references need to be cited, and some references need to be added. As in the following article: Wu, Q., Sheng, Y., Yu, Q., Chen, J., and Ma, W.: Engineering in the rugged permafrost terrain on the roof of the world under a warming climate, Permafrost and Periglacial Processes, 31, 417-428, https://doi.org/10.1002/ppp.2059, 2020.

5. This manuscript focuses on ground and drone monitoring data, so it is recommended to delete InSAR data.

---

## Author Comment (AC1) · 14 Apr 2021

Response to referee comment 1:

Dear authors, This paper gives an overview of measurement data derived from permafrost study sites in the Kunlun Mountain Pass area of the Qinghai-Tibet Plateau, China. The paper describes the locality with focus on the collocated engineered structures of the Qinghai-Tibet highway, railway and power lines. The paper is a companion to data and processing code published on zenodo.org (Meteo/ground measurements, TLS, UAV images). This paper supersedes further publications by the authors that are based in part of this data. It is highly appreciated that the authors take the extra effort to collate and describe multiple datasets into one common format and data publication. However, in the present form, the paper is incomplete w.r.t. to a number of details, the metadata describing the data as well as the processing code provided. Two datasets mentioned (Xidatan weather, ground observations, sentinel InSAR data) are not provided. Apart from textual issues I will elaborate below and in the attached commented manuscript pdf file the main issue is that I was not able to run the code in conjunction with the datasets provided. Furthermore some references are missing/misleading. Some of the figures in this paper have already appeared elsewhere (other papers by the authors). Therefore they should be clearly marked as references.

Response: Thank you for the insightful comments. In revising the paper, we have carefully considered your comments and suggestions. We agree with your comments regarding the metadata, code execution, and data description, among others. To address these concerns, we have made the following modifications to the manuscript: (1) we have added README.md files for the entire dataset of the manuscript and for each data set, such as meteorological and ground observations, TLS measurements, UAV RGB and TIR images, and R code of permafrost indices and visualization, and generated the corresponding README pdf and html files; (2) we have checked the integrity of the data file and added the missing data, including InSAR data and the study area boundary shapefile data in the TLS measurement dataset; (3) we have added vector and raster data of the boundary, DSM (digital surface model), and mosaic of the study area processed by UAV monitoring data; (4) we have renamed some data files because it was difficult for data users to obtain certain data due to naming reasons, and reorganized the file directory, (5) we have modified many inappropriate expressions, including the title; (6) we have updated the data DOI; (7) we have deleted some references with little relevance and added some related references; and (8) we have

improved the flow of the language throughout the manuscript (Figure R1). We have tried our best to address each of your points in detail. We feel the revision represents an improvement, and we hope that you agree. For more details, please see our replies below.

Specific comments:

You are using the term "hydrological-thermal deformation dynamics" and "hydrological thermal deformation" interchangeably. I understand the first term with dynamics, but am not sure the second is correct. What exactly is a hydrological deformation? I understand thermal deformation (contracting/expansion of a material under thermal stress) and I think I know what you want to say. I would rather talk about permafrost or ground dynamics or ground deformation in the context of landslides or precursor patterns rather than combining process origin (thermal/hydrological) with the observed effect (deformation) in one long term.

Response: The water, heat, and deformation of the permafrost slopes and their surroundings are monitored. We mainly want to describe the three main monitoring factors of water, heat, and deformation. For a clearer description, we have revised the title of the manuscript to "An integrated observation dataset of the hydrological-thermal deformation in permafrost slopes and engineering infrastructure in the Qinghai-Tibet Engineering Corridor". Simultaneously, for consistency of expression, the term "hydrological-thermal-deformation" is used throughout the manuscript.

Much of your intro argumentation centers around the impact of engineering structures (man made interference through the immediate built environment) on permafrost in QTP and resulting hazards. While this is clearly an important issue a number of references given do not relate to this or should be explained in a different context (see annotations). Also in your data description it is not clear what data are influenced by engineering (and possibly how much) and what data are not influenced by QTH/QTR etc.

Response: Roads, railways, and electric towers stand beside or on these two slopes. The operation of these projects affects the water, heat, and deformation of these two slopes. Therefore, the slope-related data we observe are all affected by the project operations. We have checked all the references and explained the data. For updates on some references, please see the answers below.

The data should be described concisely with correct metadata. Your data packages and references to the data in the paper do not match, file/directory naming is not explanatory. Please provide a global inventory of the data provided and exact file descriptors. Also it seems your dataset covers data from 1955-2020 (in some parts) but your paper mentions the period 2014-2019. Please clarify. Most importantly the files for ground observations are missing.

Response: We have added metadata files README.md for all data sets and generated the corresponding html and pdf format files. The study area embeds Google Maps in the README.md file. Meteorological and ground observations, as well as the R code of permafrost indices and visualization, include the period from 1955 to 2019. TLS measurements and UAV RGB and TIR images are from 2014 to 2017. We have added a description of the time period in the main text and README.md.

You are using a time-domain reflectometer (TDR) probe (model CS615-L, Campbell Scientific) for assessing the soil volumetric content. The probe is specified by the vendor (followup product CS616) for operation in 0-70âĐČ only. However you present data in figure 4 down to -16âĐČ. https://www.campbellsci.com/cs616-reflectometer Operating Temperature Range 0âĐČ to +70âĐČ. Furthermore Or, Dani, and Jon M Wraith. 1999. "Temperature Effects on Soil Bulk Dielectric Permittivity Measured by Time Domain Reflectometry: A Physical Model." Water Resources Research 35 (2): 371–83. https://doi.org/10.1029/1998WR900008. P. Overduin, Pier & Yoshikawa, Kenji & Kane, D. & Harden, J. (2005). Comparing electronic probes for volumetric water content of low-density feathermoss. Sensor Review. 25. 215-221. 10.1108/02602280510606507. Detail that it is not at all straightforward to measure

these quantities in the frozen state. Therefore I suggest to (1) remove moisture data below T = 0 ℃ or (2) at least mention that this data must be treated with utmost care as it is outside the spec of the instrument you are using.

Response: Thank you for the insightful comments. Calibrations of TDR derived from unfrozen soil may not apply to frozen soil, where water is replaced by ice (Spaans and Baker, 1995). Indeed, soil moisture below 0% is difficult to measure, and there are many uncertainties for measuring data below 0 °C. Because there are too many soil moisture data for soil temperatures below 0 °C, we have retained these data, but we have added an explanation in the text.

Soil moisture with a soil temperature below 0 °C is beyond the scope of instrument monitoring. Monitoring soil moisture under frozen conditions has always been a technical difficulty. Therefore, soil moisture data below 0 °C are not available.

Figure 4 should be labeled correctly.

Response: The Figure has been labeled correctly. Please refer to the new Figure 4 as follows:

Figure 4. Soil temperature and volumetric water content from 2014 to 2019. (a) Soil temperature (°C); (b) soil moisture (%).

You mention: "This study analyzes the thermal impact of engineering operations 240 on permafrost slopes. The results show that the QTH has the greatest thermal impact on permafrost slopes, followed by the QTR and finally the power/communication towers." I can see one figure. But where is the analysis, how is it performed and what is the quantitative outcome? Response: This was due to the lack of clarity of our expression. Based on these data, the thermal impact of different project operations on permafrost slopes was analyzed, using mostly the inflection point analysis method of regional analysis.

This study analyzes the thermal impact of engineering operations on permafrost
slopes. The projects and the slope were divided according to a width of 2 m, and then the surface temperature of the project and the temperature between different zones of the surrounding slope were compared. When these temperature differences appear at the first break point, this is the largest thermal impact of the project on the slope. The distance between the slope zone and the project is the maximum range of thermal influence (Luo et al., 2018a). The results show that the QTH has the greatest thermal impact on permafrost slopes, followed by the QTR and finally the power/communication towers.

The R code provided cannot be used. Please provide comments/readme and explain the filenames used/origin of the input files. E.g. this file referenced in the code is not available: xdt <- read.csv("PLOT/XDTMS2014-2018_PLOT.csv", header = TRUE)

Response: Thank you for the insightful comments. We have reorganized the code, added the required comments and instructions to the code, added a new instruction document on how to use the code, and added the README.md markdown file for operation of the code, including the corresponding html and pdf files. We have also recorded an operation video and provided it in README.md and README.html.

Please also note the supplement to this comment: https://essd.copernicus.org/preprints/essd-2020-106/essd-2020-106-RC1-supplement.pdf

Response: We have moved the reviewer's comments here from the manuscript edits.

L23 & 27 are the sensors "between the slopes and engineering projects" or "on and around the slopse" ?

Response: To describe the deployment of the instrument more clearly, we have deleted the sentence "and the aforementioned sensors are densely located on and around the permafrost slopes". The soil moisture sensor is deployed on the slope, the GNSS is deployed around the slope and 30 km away from the slope, the TLS performs mobile

monitoring on the slope, and the drone flies on the slope according to the planned route. The slope here also includes the projects surrounding and standing on the slope.

L37-39 yes. but is this general sentence really necessary?

Response: This has been deleted. Thank you.

L50 these references do not talk about statistical evidence for "permafrost disasters". i am sure there are better ones.

Response: We have updated the relevant references (Huggel et al., 2010;Streletskiy et al., 2019;Bessette-Kirton and Coe, 2020;Patton et al., 2019).

L51 this reference shows some nice landslide features in QTP but none of them are documented w.r.t. direct impact on engineering structures (they are somewhat close to QTH)

Response: We have updated the relevant references (Ma et al., 2006;Guo and Sun, 2015;Yu et al., 2020).

L54 see above

Response: We have updated the relevant references (Niu et al., 2015;Wirz et al., 2015).

L56 this reference is just about climate change and warming. not about man-made impact due to engineering in permafrost regions

Response: Thank you for the insightful comments. We apologize for the inappropriate references. We have updated the references (Zhang et al., 2020;Liu et al., 2020;Zhao et al., 2020).

L62 MAAT is air temperature, not soil temperature. please correct this sentence. later in the sentence it is correct Response: We apologize for the unclear expression in the

previous text. In fact, we want to express the changes in MAAT in seasonal frozen soil areas, island permafrost areas, and continuous permafrost areas, so we have rewritten this sentence.

In the past 60 years, the MAAT of the seasonal and island permafrost areas along the QTEC has increased 0.3 to 0.5 °C, and the MAAT in the continuous permafrost area has increased 0.1 to 0.3 °C (Obu et al., 2019;Luo et al., 2018b;Wu et al., 2007).

L68 this paper does not talk about deformation/destruction of engineering facilities. just about deformation of part of your data.

Response: We have updated the relevant references (Streletskiy et al., 2019;Yu et al., 2020;Ma et al., 2017).

L68-74 this section should be rewritten for clarity

Response: For clarity, we have rewritten the sentence as follows:

Warming of the climate and operation of permafrost projects around slopes have caused the ground temperature to rise. On ice-rich slopes, melting underground ice due to rising temperatures reduces the cohesion and angle of internal friction between the active layer and underground ice and becomes extremely unstable under the influence of gravity (Yuan et al., 2017). The locations of these slopes near permafrost engineering projects, such as railways and highways, thaw slumps, frost heaves, landslides, rockfalls, etc., may cause serious damage to permafrost engineering (Niu et al., 2015;Luo et al., 2018a).

L80-84 yes indeed, but can you also give some data/evidence/reference of engineered structures that are monitored and/or their susceptibility?

Response: Most of them focus on the interaction between a single project and the slope, and few studies have addressed the interaction of water-heat and deformation between multiple projects, such as highways, railways, and electric towers and slopes. In addition, the capacity of the Qinghai-Tibet Engineering Corridor (QTEC) to accommodate several infrastructure projects, such as the Qinghai-Tibet Highway, Railway, the Golmud–Lhasa Oil Pipeline, the Qinghai–Tibet Power Transmission Line, and the future Qinghai-Tibet Express Highway, must be considered. Due to severe topographical, geographical, and geological restrictions, the width of the QTEC varies from 100 m to 10 km. As a result, the mutual thermal influence of these infrastructures within the narrow corridor cannot be ignored. We have added some references (Wang et al., 2020;Ma et al., 2019).

L85 how do you do TLS _WITH_GNSS? TLS is laser reflection that is sent out by an instrument. this is often georeferenced by GNSS. please be precise.

Response: We have added the following sentence:

GNSS can be used as the datum point and control point of TLS, helping TLS point cloud data establish a georeferenced coordinate system and improving the accuracy of comparative analysis of multiple TLS data.

L88-90 you mention "visible light images" and ground surface temperature? how does this fit together?

Response: The coordinated use of multiple sensors is to obtain information from multiple angles of the two permafrost slopes from topography and landform to temperature changes. For clarity, we have rewritten the sentence as follows:

Unmanned aerial vehicles (UAVs) can be equipped with visible digital, thermal infrared (TIR), and multispectral sensors. In addition to obtaining the topographic and landform features of the two frozen soil slopes, it can also estimate the spatial distribution of the ground surface temperature on permafrost slopes and evaluate the thermal influence of nearby engineering infrastructure (Luo et al., 2018a).

L93-94 i only see data on slopes that are adjacent to engineering structures. but the data are not truly specific to the engineering structures. please reword.

Response: We have rewritten this sentence.

We provide an integrated dataset of the hydrological-thermal deformation covering permafrost engineering and slope areas in the QTEC from 2014 to 2019.

L108-109 can you please comment on the presence of permafrost? is it all permafrost? is it continuous/discontinuous? if yes/ where/how?

Response: The area around the Kunlun Mountain Pass on the Qinghai-Tibet Plateau is characterized by continuous permafrost (Figure R2). The data come from the Map of Geocryological Regionalization and Classification in China (Qiu et al., 2000).

Figure R2 The frozen soil distribution in the study area.

L115 if permafrost is only above 4200 masl, then where is it 92m? is that the maximum on QTP? or a mean? please be precise.

Response: The development of permafrost is different from that of the permafrost thickness. Permafrost is well developed at the Kunlun Mountain Pass. The permafrost thickness is the thickness of the frozen soil layer, which is derived from data from boreholes. Data from boreholes in this area show that the thickness of permafrost ranges from 46 to 112 m. The borehole data closest to the study area show that the permafrost thickness is 92 meters. To avoid misunderstanding, we have changed "92 meters" to "from 46 to 112 m".

L123 drilling or pit? where was the drilling performed? location?

Response: In 2010, two deep boreholes (please see Figure R3) were created by our institute around two slopes, which were 200 and 300 m deep (Yang et al., 2011;Yang et al., 2017;Wu and Zhang, 2008). The "depth of underground ice" is the mean depth of underground ice in the study area. "Top of permafrost" is the temperature at the top of permafrost or the temperature at the bottom of the active layer (TTOP), which differ from year to year. The warming of permafrost has become common in this region, so we have added references. We have rewritten the sentence.

Figure R3. The location of two boreholes (K1 & K2). Base map came from Yang et al.

(2017).

L130-131 the data supplements cite 2014-2020 for TLS and ground data is 2014-2018.

Response: We have added a reference.

L175 cannot be used safely below 0℃

Response: Please see our responses "2" in the Specific comments section (above).

Figure 4 no labels in the figure

Response: We have added the labels in Figure 4.

L208-209 where is this sentinel data?

Response: We have added Sentinel data to the TLS measurement data depository at http://doi.org/10.5281/zenodo.3764502

L236-237 some flights are not "from moring to afternoon". please be more specific here. also list which files are form which flight/times/area

Response: We have revised this sentence.

The TIR flight experiments lasted from morning to afternoon, with intervals of 1 to 2 hours (Table 2)

L240-242 this is no surprise. but where is the analysis and how much is the impact? please add details.

Response: Please refer to our responses "2" in the Specific comments section (above).

L256-257 how?

Response: The data were first checked manually to identify suspicious and incorrect data. Quality control codes for the meteorological station data were adopted to examine and correct the suspicious and incorrect data.

The meteorological data have undergone quality control. First, all suspicious and incorrect data were manually re-examined and corrected. For example, a new column of "Corrected_P" has been added to the precipitation data based on the original data, and this column of data is the result of manual revision.

L277-278 how? why?

Response: Trampling by wild animals may affect the deformation.

Figure B3 I cannot see any value in this figure

Response: We have deleted Figure B3.

References: Bessette-Kirton, E. K., and Coe, J. A.: A 36-Year Record of Rock Avalanches in the Saint Elias Mountains of Alaska, With Implications for Future Hazards, Frontiers in Earth Science, 8, https://doi.org/10.3389/feart.2020.00293, 2020. Guo, D., and Sun, J.: Permafrost Thaw and Associated Settlement Hazard Onset Timing over the Qinghai-Tibet Engineering Corridor, International Journal of Disaster Risk Science, 6, 347-358, https://doi.org/10.1007/s13753-015-0072-3, 2015. Huggel, C., Salzmann, N., Allen, S., Caplan-Auerbach, J., Fischer, L., Haeberli, W., Larsen, C., Schneider, D., and Wessels, R.: Recent and future warm extreme events and high-mountain slope stability, Philosophical Transactions of the Royal Society A: Mathematical, Physical and Engineering Sciences, 368, 2435-2459, https://doi.org/10.1098/rsta.2010.0078, 2010. Liu, G., Xie, C., Zhao, L., Xiao, Y., Wu, T., Wang, W., and Liu, W.: Permafrost warming near the northern limit of permafrost on the Qinghai–Tibetan Plateau during the period from 2005 to 2017: A case study in the Xidatan area, Permafrost and Periglacial Processes, https://doi.org/10.1002/ppp.2089, 2020. Luo, L., Ma, W., Zhao, W., Zhuang, Y., Zhang, Z., Zhang, M., Ma, D., and Zhou, Q.: UAV-based spatiotemporal thermal patterns of permafrost slopes along the Qinghai–Tibet Engineering Corridor, Landslides, 15, 2161–2172, https://doi.org/10.1007/s10346-018-1028-7, 2018a. Luo, L., Zhang, Z., Ma, W., Yi, S., and Zhuang, Y.: PIC v1.3: comprehensive R package for computing permafrost indices with daily weather observations and atmospheric forcing over the Qinghai–Tibet Plateau, Geosci Model Dev, 11, 2475-2491, https://doi.org/10.5194/gmd-11-2475-2018, 2018b. Ma, W., Niu, F., Akagawa, S., and Jin, D.: Slope instability phenomena in permafrost regions of Qinghai-Tibet Plateau, China, Landslides, 3, 260-264, https://doi.org/10.1007/s10346-006-0045-0, 2006. Ma, W., Mu, Y., Zhang, J., Yu, W., Zhou, Z., and Chen, T.: Lateral thermal influences of roadway and railway embankments in permafrost zones along the Qinghai-Tibet Engineering Corridor, Transportation Geotechnics, 21, https://doi.org/10.1016/j.trgeo.2019.100285, 2019. Niu, F., Luo, J., Lin, Z., Fang, J., and Liu, M.: Thaw-induced slope failures and stability analyses in permafrost regions of the Qinghai-Tibet Plateau, China, Landslides, 13, 55-65, https://doi.org/10.1007/s10346-014-0545-2, 2015. Obu, J., Westermann, S., Bartsch, A., Berdnikov, N., Christiansen, H. H., Dashtseren, A., Delaloye, R., Elberling, B., Etzelmüller, B., Kholodov, A., Khomutov, A., Kääb, A., Leibman, M. O., Lewkowicz, A. G., Panda, S. K., Romanovsky, V., Way, R. G., Westergaard-Nielsen, A., Wu, T., Yamkhin, J., and Zou, D.: Northern Hemisphere permafrost map based on TTOP modelling for 2000–2016 at 1 km2 scale, Earth-Science Reviews, 193, 299-316, https://doi.org/10.1016/j.earscirev.2019.04.023, 2019. Patton, A. I., Rathburn, S. L., and Capps, D. M.: Landslide response to climate change in permafrost regions, Geomorphology, 340, 116-128, https://doi.org/10.1016/j.geomorph.2019.04.029, 2019. Qiu, G., Zhou, Y., Guo, D., and Wang, Y.: The map of geocryological regionalization and classification in China, Science Press, Beijing (in Chinese), 2000. Spaans, E. J. A., and Baker, J. M.: Examining the use of time domain reflectometry for measuring liquid water content in frozen soil, Water Resour Res, 31, 2917-2925, https://doi.org/10.1029/95wr02769, 1995. Streletskiy, D. A., Suter, L. J., Shiklomanov, N. I., Porfiriev, B. N., and Eliseev, D. O.: Assessment of climate change impacts on buildings, structures and infrastructure in the Russian regions on permafrost, Environ Res Lett, 14, https://doi.org/10.1088/1748-9326/aaf5e6, 2019. Wang, S., Niu, F., Chen, J., and Dong, Y.: Permafrost research in China related to express highway construction, Permafrost and Periglacial Processes, 31, 406-416, https://doi.org/10.1002/ppp.2053,

2020. Wirz, V., Geertsema, M., Gruber, S., and Purves, R. S.: Temporal variability of diverse mountain permafrost slope movements derived from multi-year daily GPS data, Mattertal, Switzerland, Landslides, 13, 67-83, https://doi.org/10.1007/s10346-014-0544-3, 2015. Wu, Q., Dong, X., Liu, Y., and Jin, H.: Responses of Permafrost on the Qinghai-Tibet Plateau, China, to Climate Change and Engineering Construction, Arctic, Antarctic, and Alpine Research, 39, 682-687, https://doi.org/10.1657/1523-0430(07-508)[wu]2.0.Co;2, 2007. Wu, Q., and Zhang, T.: Recent permafrost warming on the Qinghai-Tibetan Plateau, Journal of Geophysical Research, 113, https://doi.org/10.1029/2007jd009539, 2008. Yang, Y.-z., Wu, Q.-b., Deng, Y.-s., Jiang, G.-l., and Zhang, P.: Chemical Composition of Borehole Gas in Kunlun Pass Basin in Permafrost Regions in Qinghai-Tibet Plateau, Natural Gas Geoscience, 6, 2011. Yang, Y., Wu, Q., Jiang, G., and Zhang, P.: Stable Isotopic Stratification and Growth Patterns of Ground Ice in Permafrost on the Qinghai-Tibet Plateau, China, Permafrost and Periglacial Processes, 28, 119-129, https://doi.org/10.1002/ppp.1892, 2017. Yu, W., Zhang, T., Lu, Y., Han, F., Zhou, Y., and Hu, D.: Engineering risk analysis in cold regions: State of the art and perspectives, Cold Regions Science and Technology, 171, https://doi.org/10.1016/j.coldregions.2019.102963, 2020. Yuan, C., Yu, Q., You, Y., and Guo, L.: Deformation mechanism of an expressway embankment in warm and high ice content permafrost regions, Appl Therm Eng, 121, 1032-1039, https://doi.org/10.1016/j.applthermaleng.2017.04.128, 2017. Zhang, Z., Yu, Q., You, Y., Guo, L., Wang, X., Liu, G., and Wu, G.: Cooling effect analysis of temperature-controlled ventilated embankment in Qinghai-Tibet testing expressway, Cold Regions Science and Technology, 173, https://doi.org/10.1016/j.coldregions.2020.103012, 2020. Zhao, L., Zou, D., Hu, G., Du, E., Pang, Q., Xiao, Y., Li, R., Sheng, Y., Wu, X., Sun, Z., Wang, L., Wang, C., Ma, L., Zhou, H., and Liu, S.: Changing climate and the permafrost environment on the Qinghai–Tibet (Xizang) plateau, Permafrost and Periglacial Processes, 31, 396-405, https://doi.org/10.1002/ppp.2056, 2020.

Please also note the supplement to this comment:

https://essd.copernicus.org/preprints/essd-2020-106/essd-2020-106-AC1-supplement.pdf

**Supplement:**

**"An integrated observation dataset of the hydrological-thermal-deformation dynamics in the permafrost slopes and engineering infrastructure in the Qinghai-Tibet Engineering Corridor"**

**by Lihui Luo et al.**

We thank Dr. Jan Beutel for valuable feedback, which helped us improve the manuscript. Please find below the Referee comments in black, Author responses in green, and Changes to the manuscript in blue.

**Response to referee comment 1:**

Dear authors,

This paper gives an overview of measurement data derived from permafrost study sites in the Kunlun Mountain Pass area of the Qinghai-Tibet Plateau, China. The paper describes the locality with focus on the collocated engineered structures of the Qinghai-Tibet highway, railway and power lines. The paper is a companion to data and processing code published on zenodo.org (Meteo/ground measurements, TLS, UAV images). This paper supersedes further publications by the authors that are based in part of this data. It is highly appreciated that the authors take the extra effort to collate and describe multiple datasets into one common format and data publication. However, in the present form, the paper is incomplete w.r.t. to a number of details, the metadata describing the data as well as the processing code provided. Two datasets mentioned (Xidatan weather, ground observations, sentinel InSAR data) are not provided. Apart from textual issues I will elaborate below and in the attached commented manuscript pdf file the main issue is that I was not able to run the code in conjunction with the datasets provided. Furthermore some references are missing/misleading. Some of the figures in this paper have already appeared elsewhere

(other papers by the authors). Therefore they should be clearly marked as references.

Thank you for the insightful comments. In revising the paper, we have carefully considered your comments and suggestions. We agree with your comments regarding the metadata, code execution, and data description, among others. To address these concerns, we have made the following modifications to the manuscript: (1) we have added README.md files for the entire dataset of the manuscript and for each data set, such as meteorological and ground observations, TLS measurements, UAV RGB and TIR images, and R code of permafrost indices and visualization, and generated the corresponding README pdf and html files; (2) we have checked the integrity of the data file and added the missing data, including InSAR data and the study area boundary shapefile data in the TLS measurement dataset; (3) we have added vector and raster data of the boundary, DSM (digital surface model), and mosaic of the study area processed by UAV monitoring data; (4) we have renamed some data files because it was difficult for data users to obtain certain data due to naming reasons, and reorganized the file directory, (5) we have modified many inappropriate expressions, including the title; (6) we have updated the data DOI; (7) we have deleted some references with little relevance and added some related references; and (8) we have improved the flow of the language throughout the manuscript (Figure R1). We have tried our best to address each of your points in detail. We feel the revision represents an improvement, and we hope that you agree. For more details, please see our replies below.

[Figure]

[Figure]

Figure R1. Editorial Certificate.

Specific comments:

You are using the term "hydrological-thermal deformation dynamics" and "hydrological thermal deformation" interchangeably. I understand the first term with dynamics, but am not sure the second is correct. What exactly is a hydrological deformation? I understand thermal deformation (contracting/expansion of a material under thermal stress) and I think I know what you want to say. I

10    would rather talk about permafrost or ground dynamics or ground deformation in the context of landslides or precursor patterns rather than combining process origin (thermal/hydrological) with the observed effect (deformation) in one long term.

The water, heat, and deformation of the permafrost slopes and their surroundings are monitored. We mainly want to describe the three main monitoring factors of water, heat, and deformation. For a clearer

15    description, we have revised the title of the manuscript to "An integrated observation dataset of the hydrological-thermal deformation in permafrost slopes and engineering infrastructure in the Qinghai-Tibet Engineering Corridor". Simultaneously, for consistency of expression, the term "hydrologicalthermal-deformation" is used throughout the manuscript.

Much of your intro argumentation centers around the impact of engineering structures (man made interference through the immediate built environment) on permafrost in QTP and resulting hazards. While this is clearly an important issue a number of references given do not relate to this or should be explained in a different context (see annotations). Also in your data description it is not clear what data are influenced by engineering (and possibly how much) and what data are not influenced by QTH/QTR etc.

Roads, railways, and electric towers stand beside or on these two slopes. The operation of these projects affects the water, heat, and deformation of these two slopes. Therefore, the slope-related data we observe are all affected by the project operations. We have checked all the references and explained the data. For updates on some references, please see the answers below.

The data should be described concisely with correct metadata. Your data packages and references to the data in the paper do not match, file/directory naming is not explanatory. Please provide a global inventory of the data provided and exact file descriptors. Also it seems your dataset covers data from 1955-2020 (in some parts) but your paper mentions the period 2014-2019. Please clarify. Most importantly the files for ground observations are missing.

We have added metadata files README.md for all data sets and generated the corresponding html and pdf format files. The study area embeds Google Maps in the README.md file. Meteorological and ground observations, as well as the R code of permafrost indices and visualization, include the period from 1955 to 2019. TLS measurements and UAV RGB and TIR images are from 2014 to 2017. We have added a description of the time period in the main text and README.md.

You are using a time-domain reflectometer (TDR) probe (model CS615-L, Campbell Scientific) for assessing the soil volumetric content. The probe is specified by the vendor (followup product CS616) for operation in 0-70℃ only. However you present data in figure 4 down to -16℃. https://www.campbellsci.com/cs616-reflectometer Operating Temperature Range 0℃ to +70℃. Furthermore Or, Dani, and Jon M Wraith. 1999. "Temperature Effects on Soil Bulk Dielectric Permittivity Measured by Time Domain Reflectometry: A Physical Model." Water Resources Research

35 (2): 371–83. https://doi.org/10.1029/1998WR900008. P. Overduin, Pier & Yoshikawa, Kenji & Kane,

D. & Harden, J. (2005). Comparing electronic probes for volumetric water content of low-density

feathermoss. Sensor Review. 25. 215-221. 10.1108/02602280510606507. Detail that it is not at all

straightforward to measure these quantities in the frozen state. Therefore I suggest to (1) remove moisture

5    data below T = 0 °C or (2) at least mention that this data must be treated with utmost care as it is outside

the spec of the instrument you are using.

Thank you for the insightful comments. Calibrations of TDR derived from unfrozen soil may not apply

to frozen soil, where water is replaced by ice (Spaans and Baker, 1995). Indeed, soil moisture below 0%

is difficult to measure, and there are many uncertainties for measuring data below 0 °C. Because there

10    are too many soil moisture data for soil temperatures below 0 °C, we have retained these data, but we

have added an explanation in the text.

Soil moisture with a soil temperature below 0 °C is beyond the scope of instrument monitoring.

Monitoring soil moisture under frozen conditions has always been a technical difficulty. Therefore, soil

15    moisture data below 0 °C are not available.

Figure 4 should be labeled correctly.

The Figure has been labeled correctly. Please refer to the new Figure 4 as follows:

[Figure]

20    Figure 4. Soil temperature and volumetric water content from 2014 to 2019. (a) Soil temperature (°C);

(b) soil moisture (%).

You mention: "This study analyzes the thermal impact of engineering operations 240 on permafrost slopes. The results show that the QTH has the greatest thermal impact on permafrost slopes, followed by the QTR and finally the power/communication towers." I can see one figure. But where is the analysis, how is it performed and what is the quantitative outcome?

5    This was due to the lack of clarity of our expression. Based on these data, the thermal impact of different project operations on permafrost slopes was analyzed, using mostly the inflection point analysis method of regional analysis.

This study analyzes the thermal impact of engineering operations on permafrost slopes. The projects and

10    the slope were divided according to a width of 2 m, and then the surface temperature of the project and the temperature between different zones of the surrounding slope were compared. When these temperature differences appear at the first break point, this is the largest thermal impact of the project on the slope. The distance between the slope zone and the project is the maximum range of thermal influence (Luo et al., 2018a). The results show that the QTH has the greatest thermal impact on permafrost slopes,

15    followed by the QTR and finally the power/communication towers.

The R code provided cannot be used. Please provide comments/readme and explain the filenames used/origin of the input files. E.g. this file referenced in the code is not available: xdt <-read.csv("PLOT/XDTMS2014-2018_PLOT.csv", header = TRUE)

20    Thank you for the insightful comments. We have reorganized the code, added the required comments and instructions to the code, added a new instruction document on how to use the code, and added the README.md markdown file for operation of the code, including the corresponding html and pdf files. We have also recorded an operation video and provided it in README.md and README.html.

25    Please also note the supplement to this comment:

https://essd.copernicus.org/preprints/essd-2020-106/essd-2020-106-RC1-supplement.pdf

We have moved the reviewer's comments here from the manuscript edits.

L23 & 27 are the sensors "between the slopes and engineering projects" or "on and around the slopse" ?

30    To describe the deployment of the instrument more clearly, we have deleted the sentence "and the

aforementioned sensors are densely located on and around the permafrost slopes". The soil moisture sensor is deployed on the slope, the GNSS is deployed around the slope and 30 km away from the slope, the TLS performs mobile monitoring on the slope, and the drone flies on the slope according to the planned route. The slope here also includes the projects surrounding and standing on the slope.

L37-39 yes. but is this general sentence really necessary?

This has been deleted. Thank you.

L50 these references do not talk about statistical evidence for "permafrost disasters". i am sure there are better ones.

We have updated the relevant references (Huggel et al., 2010;Streletskiy et al., 2019;Bessette-Kirton and Coe, 2020;Patton et al., 2019).

L51 this reference shows some nice landslide features in QTP but none of them are documented w.r.t. direct impact on engineering structures (they are somewhat close to QTH)

We have updated the relevant references (Ma et al., 2006;Guo and Sun, 2015;Yu et al., 2020).

L54 see above

We have updated the relevant references (Niu et al., 2015;Wirz et al., 2015).

L56 this reference is just about climate change and warming. not about man-made impact due to engineering in permafrost regions

Thank you for the insightful comments. We apologize for the inappropriate references. We have updated the references (Zhang et al., 2020;Liu et al., 2020;Zhao et al., 2020).

L62 MAAT is air temperature, not soil temperature. please correct this sentence. later in the sentence it is correct

We apologize for the unclear expression in the previous text. In fact, we want to express the changes in MAAT in seasonal frozen soil areas, island permafrost areas, and continuous permafrost areas, so we have rewritten this sentence.

In the past 60 years, the MAAT of the seasonal and island permafrost areas along the QTEC has increased 0.3 to 0.5 °C, and the MAAT in the continuous permafrost area has increased 0.1 to 0.3 °C (Obu et al., 2019;Luo et al., 2018b;Wu et al., 2007).

L68 this paper does not talk about deformation/destruction of engineering facilities. just about deformation of part of your data.

We have updated the relevant references (Streletskiy et al., 2019;Yu et al., 2020;Ma et al., 2017).

10     L68-74 this section should be rewritten for clarity

For clarity, we have rewritten the sentence as follows:

Warming of the climate and operation of permafrost projects around slopes have caused the ground temperature to rise. On ice-rich slopes, melting underground ice due to rising temperatures reduces the

15     cohesion and angle of internal friction between the active layer and underground ice and becomes extremely unstable under the influence of gravity (Yuan et al., 2017). The locations of these slopes near permafrost engineering projects, such as railways and highways, thaw slumps, frost heaves, landslides, rockfalls, etc., may cause serious damage to permafrost engineering (Niu et al., 2015;Luo et al., 2018a).

20     L80-84 yes indeed, but can you also give some data/evidence/reference of engineered structures that are monitored and/or their susceptibility?

Most of them focus on the interaction between a single project and the slope, and few studies have addressed the interaction of water-heat and deformation between multiple projects, such as highways, railways, and electric towers and slopes. In addition, the capacity of the Qinghai-Tibet Engineering

25     Corridor (QTEC) to accommodate several infrastructure projects, such as the Qinghai-Tibet Highway, Railway, the Golmud–Lhasa Oil Pipeline, the Qinghai–Tibet Power Transmission Line, and the future Qinghai-Tibet Express Highway, must be considered. Due to severe topographical, geographical, and geological restrictions, the width of the QTEC varies from 100 m to 10 km. As a result, the mutual thermal influence of these infrastructures within the narrow corridor cannot be ignored. We have added

30     some references (Wang et al., 2020;Ma et al., 2019).

L85 how do you do TLS _WITH_GNSS? TLS is laser reflection that is sent out by an instrument. this is often georeferenced by GNSS. please be precise.

We have added the following sentence:

GNSS can be used as the datum point and control point of TLS, helping TLS point cloud data establish a georeferenced coordinate system and improving the accuracy of comparative analysis of multiple TLS data.

L88-90 you mention "visible light images" and ground surface temperature? how does this fit together?

The coordinated use of multiple sensors is to obtain information from multiple angles of the two permafrost slopes from topography and landform to temperature changes. For clarity, we have rewritten the sentence as follows:

Unmanned aerial vehicles (UAVs) can be equipped with visible digital, thermal infrared (TIR), and multispectral sensors. In addition to obtaining the topographic and landform features of the two frozen soil slopes, it can also estimate the spatial distribution of the ground surface temperature on permafrost slopes and evaluate the thermal influence of nearby engineering infrastructure (Luo et al., 2018a).

L93-94 i only see data on slopes that are adjacent to engineering structures. but the data are not truly specific to the engineering structures. please reword.

We have rewritten this sentence.

We provide an integrated dataset of the hydrological-thermal deformation covering permafrost engineering and slope areas in the QTEC from 2014 to 2019.

L108-109 can you please comment on the presence of permafrost? is it all permafrost? is it continuous/discontinuous? if yes/ where/how?

The area around the Kunlun Mountain Pass on the Qinghai-Tibet Plateau is characterized by continuous permafrost (Figure R2). The data come from the Map of Geocryological Regionalization and

Classification in China (Qiu et al., 2000).

[Figure]

Figure R2 The frozen soil distribution in the study area.

L115 if permafrost is only above 4200 masl, then where is it 92m? is that the maximum on QTP? or a mean? please be precise.

The development of permafrost is different from that of the permafrost thickness. Permafrost is well developed at the Kunlun Mountain Pass. The permafrost thickness is the thickness of the frozen soil layer, which is derived from data from boreholes. Data from boreholes in this area show that the thickness of permafrost ranges from 46 to 112 m. The borehole data closest to the study area show that the permafrost thickness is 92 meters. To avoid misunderstanding, we have changed "92 meters" to "from 46 to 112 m".

L123 drilling or pit? where was the drilling performed? location?

In 2010, two deep boreholes (please see Figure R3) were created by our institute around two slopes, which were 200 and 300 m deep (Yang et al., 2011;Yang et al., 2017;Wu and Zhang, 2008). The "depth of underground ice" is the mean depth of underground ice in the study area. "Top of permafrost" is the temperature at the top of permafrost or the temperature at the bottom of the active layer (TTOP), which differ from year to year. The warming of permafrost has become common in this region, so we have added references. We have rewritten the sentence.

[Figure]

Figure R3. The location of two boreholes (K1 & K2). Base map came from Yang et al. (2017).

L130-131 the data supplements cite 2014-2020 for TLS and ground data is 2014-2018.

We have added a reference.

L175 cannot be used safely below 0℃

Please see our responses "2" in the Specific comments section (above).

Figure 4 no labels in the figure

We have added the labels in Figure 4.

L208-209 where is this sentinel data?

We have added Sentinel data to the TLS measurement data depository at http://doi.org/10.5281/zenodo.3764502

L236-237 some flights are not "from moring to afternoon". please be more specific here. also list which

files are form which flight/times/area

We have revised this sentence.

The TIR flight experiments lasted from morning to afternoon, with intervals of 1 to 2 hours (Table 2)

L240-242 this is no surprise. but where is the analysis and how much is the impact? please add details.

Please refer to our responses "2" in the Specific comments section (above).

L256-257 how?

10  The data were first checked manually to identify suspicious and incorrect data. Quality control codes for the meteorological station data were adopted to examine and correct the suspicious and incorrect data.

The meteorological data have undergone quality control. First, all suspicious and incorrect data were manually re-examined and corrected. For example, a new column of "Corrected_P" has been added to

15  the precipitation data based on the original data, and this column of data is the result of manual revision.

L277-278 how? why?

Trampling by wild animals may affect the deformation.

20  Figure B3 I cannot see any value in this figure

We have deleted Figure B3.

**References:**

Bessette-Kirton, E. K., and Coe, J. A.: A 36-Year Record of Rock Avalanches in the Saint Elias Mountains

25      of Alaska, With Implications for Future Hazards, Frontiers in Earth Science, 8, https://doi.org/10.3389/feart.2020.00293, 2020.

Guo, D., and Sun, J.: Permafrost Thaw and Associated Settlement Hazard Onset Timing over the Qinghai-Tibet Engineering Corridor, International Journal of Disaster Risk Science, 6, 347-358, https://doi.org/10.1007/s13753-015-0072-3, 2015.

30  Huggel, C., Salzmann, N., Allen, S., Caplan-Auerbach, J., Fischer, L., Haeberli, W., Larsen, C., Schneider,

D., and Wessels, R.: Recent and future warm extreme events and high-mountain slope stability, Philosophical Transactions of the Royal Society A: Mathematical, Physical and Engineering Sciences, 368, 2435-2459, https://doi.org/10.1098/rsta.2010.0078, 2010.

Liu, G., Xie, C., Zhao, L., Xiao, Y., Wu, T., Wang, W., and Liu, W.: Permafrost warming near the northern limit of permafrost on the Qinghai–Tibetan Plateau during the period from 2005 to 2017: A case study in the Xidatan area, Permafrost and Periglacial Processes, https://doi.org/10.1002/ppp.2089, 2020.

Luo, L., Ma, W., Zhao, W., Zhuang, Y., Zhang, Z., Zhang, M., Ma, D., and Zhou, Q.: UAV-based spatiotemporal thermal patterns of permafrost slopes along the Qinghai–Tibet Engineering Corridor, Landslides, 15, 2161–2172, https://doi.org/10.1007/s10346-018-1028-7, 2018a.

Luo, L., Zhang, Z., Ma, W., Yi, S., and Zhuang, Y.: PIC v1.3: comprehensive R package for computing permafrost indices with daily weather observations and atmospheric forcing over the Qinghai–Tibet Plateau, Geosci Model Dev, 11, 2475-2491, https://doi.org/10.5194/gmd-11-2475-2018, 2018b.

Ma, W., Niu, F., Akagawa, S., and Jin, D.: Slope instability phenomena in permafrost regions of Qinghai-Tibet Plateau, China, Landslides, 3, 260-264, https://doi.org/10.1007/s10346-006-0045-0, 2006.

Ma, W., Mu, Y., Zhang, J., Yu, W., Zhou, Z., and Chen, T.: Lateral thermal influences of roadway and railway embankments in permafrost zones along the Qinghai-Tibet Engineering Corridor, Transportation Geotechnics, 21, https://doi.org/10.1016/j.trgeo.2019.100285, 2019.

Niu, F., Luo, J., Lin, Z., Fang, J., and Liu, M.: Thaw-induced slope failures and stability analyses in permafrost regions of the Qinghai-Tibet Plateau, China, Landslides, 13, 55-65, https://doi.org/10.1007/s10346-014-0545-2, 2015.

Obu, J., Westermann, S., Bartsch, A., Berdnikov, N., Christiansen, H. H., Dashtseren, A., Delaloye, R., Elberling, B., Etzelmüller, B., Kholodov, A., Khomutov, A., Kääb, A., Leibman, M. O., Lewkowicz, A. G., Panda, S. K., Romanovsky, V., Way, R. G., Westergaard-Nielsen, A., Wu, T., Yamkhin, J., and Zou, D.: Northern Hemisphere permafrost map based on TTOP modelling for 2000–2016 at 1 km2 scale, Earth-Science Reviews, 193, 299-316, https://doi.org/10.1016/j.earscirev.2019.04.023, 2019.

Patton, A. I., Rathburn, S. L., and Capps, D. M.: Landslide response to climate change in permafrost regions, Geomorphology, 340, 116-128, https://doi.org/10.1016/j.geomorph.2019.04.029, 2019.

Qiu, G., Zhou, Y., Guo, D., and Wang, Y.: The map of geocryological regionalization and classification

in China, Science Press, Beijing (in Chinese), 2000.

Spaans, E. J. A., and Baker, J. M.: Examining the use of time domain reflectometry for measuring liquid water content in frozen soil, Water Resour Res, 31, 2917-2925, https://doi.org/10.1029/95wr02769, 1995.

5 Streletskiy, D. A., Suter, L. J., Shiklomanov, N. I., Porfiriev, B. N., and Eliseev, D. O.: Assessment of climate change impacts on buildings, structures and infrastructure in the Russian regions on permafrost, Environ Res Lett, 14, https://doi.org/10.1088/1748-9326/aaf5e6, 2019.

Wang, S., Niu, F., Chen, J., and Dong, Y.: Permafrost research in China related to express highway construction, Permafrost and Periglacial Processes, 31, 406-416, 10 https://doi.org/10.1002/ppp.2053, 2020.

Wirz, V., Geertsema, M., Gruber, S., and Purves, R. S.: Temporal variability of diverse mountain permafrost slope movements derived from multi-year daily GPS data, Mattertal, Switzerland, Landslides, 13, 67-83, https://doi.org/10.1007/s10346-014-0544-3, 2015.

Wu, Q., Dong, X., Liu, Y., and Jin, H.: Responses of Permafrost on the Qinghai-Tibet Plateau, China, to 15 Climate Change and Engineering Construction, Arctic, Antarctic, and Alpine Research, 39, 682-687, https://doi.org/10.1657/1523-0430(07-508)[wu]2.0.Co;2, 2007.

Wu, Q., and Zhang, T.: Recent permafrost warming on the Qinghai-Tibetan Plateau, Journal of Geophysical Research, 113, https://doi.org/10.1029/2007jd009539, 2008.

Yang, Y.-z., Wu, Q.-b., Deng, Y.-s., Jiang, G.-l., and Zhang, P.: Chemical Composition of Borehole Gas 20 in Kunlun Pass Basin in Permafrost Regions in Qinghai-Tibet Plateau, Natural Gas Geoscience, 6, 2011.

Yang, Y., Wu, Q., Jiang, G., and Zhang, P.: Stable Isotopic Stratification and Growth Patterns of Ground Ice in Permafrost on the Qinghai-Tibet Plateau, China, Permafrost and Periglacial Processes, 28, 119-129, https://doi.org/10.1002/ppp.1892, 2017.

25 Yu, W., Zhang, T., Lu, Y., Han, F., Zhou, Y., and Hu, D.: Engineering risk analysis in cold regions: State of the art and perspectives, Cold Regions Science and Technology, 171, https://doi.org/10.1016/j.coldregions.2019.102963, 2020.

Yuan, C., Yu, Q., You, Y., and Guo, L.: Deformation mechanism of an expressway embankment in warm and high ice content permafrost regions, Appl Therm Eng, 121, 1032-1039, 30 https://doi.org/10.1016/j.applthermaleng.2017.04.128, 2017.

Zhang, Z., Yu, Q., You, Y., Guo, L., Wang, X., Liu, G., and Wu, G.: Cooling effect analysis of temperature-controlled ventilated embankment in Qinghai-Tibet testing expressway, Cold Regions Science and Technology, 173, https://doi.org/10.1016/j.coldregions.2020.103012, 2020.

Zhao, L., Zou, D., Hu, G., Du, E., Pang, Q., Xiao, Y., Li, R., Sheng, Y., Wu, X., Sun, Z., Wang, L., Wang, C., Ma, L., Zhou, H., and Liu, S.: Changing climate and the permafrost environment on the Qinghai–Tibet (Xizang) plateau, Permafrost and Periglacial Processes, 31, 396-405, https://doi.org/10.1002/ppp.2056, 2020.

**Data description for essd-2020-106**

**An integrated observation dataset of the hydrological-thermal-deformation in the permafrost slopes and engineering infrastructure in the Qinghai-Tibet Engineering Corridor**

**Description**

**Meteorological observations** Observation of meteorological factors was conducted at two permanent meteorological stations (Golmud and Wudaoliang) and one field meteorological station (Xidatan) with daily meteorological records. All three meteorological stations contain ground observations.

**Ground observations** The ground temperature and moisture data from the near-surface to within 270 cm in the active layer were recorded. In situ ground observations were deployed starting in July 2013 using thermocouple probes (105T, Campbell Scientific) to measure the soil temperature and using 11 time-domain reflectometer (TDR) probes (model CS615-L, Campbell Scientific) to measure the soil volumetric water content.

**TLS measurements** A FARO Focus3D X130 3D laser scanner and six Trimble 5700 GNSS systems were deployed around permafrost slopes between May 2014 and October 2015. As a supplement to the TLS point cloud data, we used Interferometric Synthetic Aperture Radar (InSAR) technology to prepare Sentinel-1 deformation data for the study area from 2014 to 2020.

**UAV RGB and TIR images** Two permafrost slopes were conducted four flight experiments with UAV-mounted RGB and TIR sensors in 2016 and 2017.

**R code of permafrost indices and visualization** R Script for plotting meteorological observation data and permafrost indices (MAAT and MAGST) during 1955-2018.

**Keywords**

**Theme:** Permafrost slope; Permafrost engineering; Freeze-thaw; hydrological-thermal-Deformation; Qinghai-Tibet plateau

**Discipline:** cryosphere; In-situ monitoring data; Remote sensing data using TLS and UAV

**Places:** Qinghai-Tibet Engineering Corridor; Kunlun Mountain Pass close to Hoh Xil Nature Reserve

**Data details**

**Scale:** UAV RGB: ~5 cm; UAV TIR: ~ 20 cm; TLS measurements: 0.009°

**Coordinate Reference System:** EPSG: 4326 - WGS 84

**Filesize:**~ 5 G

**Data format:** GeoTiff, CSV, EXCEL XLSX, TXT, WRP, Tif, JPG

**Space scope**

```
                North: 35°39′ 10″
    West: 90°3′ 30″    –       East: 90°3′ 55″
                South: 35°38′ 35″
```

[Figure]

Map data ©2021 Imagery ©2021 CNES / Airbus, Maxar Technologies

**Time period**

Table 1. Observations period of all datasets.

| Data Type | Location | Period | Remark |
|---|---|---|---|
| Meteorological observations | Golmud station | 1955-2018 | National Reference Station |
| Meteorological observations | Xidatan station | 2014-2018 | National General Station |
| Meteorological observations | Wudaoliang station | 1956-2018 | National Reference Station |
| Ground observations | Study Area | 2014-2019 | Field test site |
| Ground observations | Golmud station | 1955-2018 | National Reference Station |

| Data Type | Location | Period | Remark |
|---|---|---|---|
| Ground observations | Xidatan station | 2014-2018 | National General Station |
| Ground observations | Wudaoliang station | 1956-2018 | National Reference Station |
| TLS measurements | Study Area | 2014-2015 | Contains measurement and comparative analysis data |
| InSAR | Study Area | 2014-2020 | Contains thawing and freezing period data |
| UAV RGB and TIR images | Study Area | 2016-2017 | tif & jpg can be processed by Pix4Dmapper & FLIR |
| R code of permafrost indices and visualization | Stations | 1955-2018 | Plot Fig. 2 & F1; Computing MAAT & MAGST |

**Meteorological and Ground observations**

**Table 2.** Observations period of datasets.

| Data Type | Location | Period | File Names |
|---|---|---|---|
| Meteorological observations | Golmud station | 1955-2018 | Meteo_52818_Golmud_1955-2010.dat;Meteo_52818_Golmud_2010-2018.xlsx |
| Meteorological observations | Xidatan station | 2014-2018 | Meteo_00000_Golmud_2014-2019.xlsx |
| Meteorological observations | Wudaoliang station | 1956-2018 | Meteo_52908_Wudaoliang_1956-2010.dat;Meteo_52908_Wudaoliang_2010-2018.xlsx |
| Ground observations | Study Area | 2014-2019 | GT00000_Slopes_2014-2019.xlsx |

| Data Type | Location | Period | File Names |
|-----------|----------|--------|------------|
| Ground observations | Golmud station | 1955-2018 | GT52818_Golmud.txt |
| Ground observations | Xidatan station | 2014-2018 | Meteo_00000_Xidatan_2014-2019.xlsx |
| Ground observations | Wudaoliang station | 1956-2018 | GT52908_Wudaoliang.txt |

**Table 3.** Ground data Metadata of meteorological stations data. The file name with **'GT'** is ground observation data.

| | ID | Variable | Type | Field Name | Unit | Description |
|---|----|----------|------|------------|------|-------------|
| 1 | 1 | Station ID | Number(5) | V01000 | | |
| 2 | 5 | Year | Number(4) | V04001 | Year | |
| 3 | 6 | Month | Number(2) | V04002 | Month | |
| 4 | 7 | Day | Number(2) | V04003 | Day | |
| 5 | 32 | Evaporation | Number(6) | V13241 | 0.1mm | evaporation |
| 6 | 53 | average ground temperature at 0 cm | Number(6) | V12240 | 0.1℃ | GT_0_AVG |
| 7 | 54 | daily maximum ground temperature at 0 cm | Number(6) | V12213 | 0.1℃ | GT_0_MAX |
| 8 | 56 | daily minimum ground temperature at 0 cm | Number(6) | V12214 | 0.1℃ | GT_0_MIN |

| | ID | Variable | Type | Field Name | Unit | Description |
|---|----|----------|------|-----------|------|-------------|
| 9 | 58 | average ground temperature at 5 cm | Number(6) | V12240_005 | 0.1℃ | GT_5_AVG |
| 10 | 59 | average ground temperature at 10 cm | Number(6) | V12240_010 | 0.1℃ | GT_10_AVG |
| 11 | 60 | average ground temperature at 15 cm | Number(6) | V12240_015 | 0.1℃ | GT_15_AVG |
| 12 | 61 | average ground temperature at 20 cm | Number(6) | V12240_020 | 0.1℃ | GT_20_AVG |
| 13 | 62 | average ground temperature at 40 cm | Number(6) | V12240_040 | 0.1℃ | GT_40_AVG |
| 14 | 63 | average ground temperature at 50 cm | Number(6) | V12240_050 | 0.1℃ | GT_50_AVG |
| 15 | 64 | average ground temperature at 80 cm | Number(6) | V12240_080 | 0.1℃ | GT_80_AVG |
| 16 | 65 | average ground temperature at 160 cm | Number(6) | V12240_160 | 0.1℃ | GT_160_AVG |
| 17 | 66 | average ground temperature at 320 cm | Number(6) | V12240_320 | 0.1℃ | GT_320_AVG |

| ID | Variable | Type | Field Name | Unit | Description |
|---|---|---|---|---|---|
| | | | | | |

Table 4. Meteorological Metadata of meteorological stations data. The file name with **'Meteo'** is Meteorological observation data.

| | ID | Variable | Type | Unit | Description |
|---|---|---|---|---|---|
| 1 | 1 | Station ID | Number(5) | | |
| 2 | 5 | Year | Number(4) | Year | Year |
| 3 | 6 | Month | Number(2) | Month | Mon |
| 4 | 7 | Day | Number(2) | Day | Day |
| 5 | 32 | daily mean air temperature at 2 m | Number(6) | 0.1℃ | Temperate |
| 6 | 53 | maximum air temperature at 2 m | Number(6) | 0.1℃ | Tmax |
| 7 | 54 | minimum air temperature at 2 m | Number(6) | 0.1℃ | Tmin |
| 8 | 56 | average wind speed | Number(6) | 0.1℃ | Wind |
| 9 | 58 | average precipitation | Number(6) | 0.1mm | Precip |
| 10 | 59 | Corrected average precipitation | Number(6) | 0.1℃ | Corrected_P |
| 11 | 60 | Evaporation | Number(6) | 0.1mm | Evaporation |
| 12 | 61 | Air humidity | Number(6) | % | Humidity |
| 13 | 62 | Air pressure | Number(6) | 0.1Pa | Press |
| 14 | 63 | sunshine time | Number(6) | 0.1h | Sunshine |
| 15 | 64 | average ground temperature at 0 cm | Number(6) | 0.1℃ | GT |

**TLS measurements**

**TLS measurements** There are a total of 4 monitorings between May 2014 and October 2015 within two thawing periods and a freezing period. The three freeze-thaw phases are referred to as "first thawing" (May 2014 to October 2014, called here "period 2-1"), "first

freezing" (October 2014 to May 2015, called here "period 3-2"), "second thawing" (May 2015 to October 2015, called here "period 4-3"), "one thawing and one freezing stage" (May 2014 to May 2015, called here "period 3-1"), and "two thawing and one freezing stage" (May 2014 to October 2015, called here "period 4-1") in the following. The file directories for each monitoring are: first, second, third, and fourth. And the file also contains comparative analysis data of different periods.

Table 5 Freeze-thaw stages of TLS scanner data.

| Status | Condition | Date Span | Days | Slope | Data points |
|--------|-----------|-----------|------|-------|-------------|
| Period 2-1 | Thawing | 05/02/2014–10/10/2014 | 161 | Slope A | 1251706 |
| Period 2-1 | Thawing | 05/02/2014–10/10/2014 | 161 | Slope B | 1367438 |
| Period 3-2 | Freezing | 10/10/2014–05/03/2015 | 205 | Slope A | 1291356 |
| Period 3-2 | Freezing | 10/10/2014–05/03/2015 | 205 | Slope B | 1366141 |
| Period 4-3 | Thawing | 05/03/2015–10/04/2015 | 154 | Slope A | 1248325 |
| Period 4-3 | Thawing | 05/03/2015–10/04/2015 | 154 | Slope B | 1382768 |
| Period 3-1 | one thawing and one freezing | 05/02/2014–05/03/2015 | 366 | Slope A | 1278448 |
| Period 3-1 | one thawing and one freezing | 05/02/2014–05/03/2015 | 366 | Slope B | 1279204 |
| Period 4-1 | two thawing and one freezing | 05/02/2014–10/04/2015 | 520 | Slope A | 1279706 |
| Period 4-1 | two thawing and one freezing | 05/02/2014–10/04/2015 | 520 | Slope B | 1207493 |

**InSAR data** The Sentinel-1 mission provides data from a dual-polarization C-band Synthetic Aperture Radar (SAR) instrument. This collection includes the S1 Ground Range Detected (GRD) scenes, processed using the Sentinel-1 Toolbox to generate a calibrated, ortho-corrected product. File directory is InSAR.

**Table 6.** InSAR data for Permafrost slope A & B, including the study area vector shapefile file(SlopeAB). Direction of the orbit ('ASCENDING' or 'DESCENDING') for the oldest image data in the product (the start of the product). The spatial resolution is 10 meters.

| Data Type | Period | Condition | Remark |
|---|---|---|---|
| asc | 2014-2016 | Tawing | ASCENDING |
| asc | 2014-2017 | Freezing | ASCENDING |
| asc | 2017-2019 | Tawing | ASCENDING |
| asc | 2017-2020 | Freezing | ASCENDING |
| desc | 2014-2016 | Tawing | DESCENDING |
| desc | 2014-2017 | Freezing | DESCENDING |
| desc | 2017-2019 | Tawing | DESCENDING |
| desc | 2017-2020 | Freezing | DESCENDING |
| Study Area boundary | | | SlopeAB:Shapefile |

**UAV RGB and TIR images**

For these two slopes, we conducted four flight experiments with UAV-mounted RGB and TIR sensors. The directory of flight images for RGB and thermal infrared sensors is RGB and TIR.

There are three directories under the RGB directory: **20160417, 20160830 and 20170822**, the format is yyyyymmdd, which represent the UAV photos taken by the RGB camera that day. Please use **exiftool** to view the metadata information of pictures such as timestamp and location.

There are three directories under the TIR directory: **2016SlopeA and 2017SlopeAB**, the format is yyyyySlope, which represent UAV photos taken by the TIR sensor of the year.

> Please use exiftool to view the metadata information of pictures such as timestamp, location, and center point temperature.

> To obtain temperatures, a sensor that is able to provide absolute temperature is needed (instead of relative temperature). The FLIR Vue Pro and the Zenmuse XT do not provide absolute temperature. However, the FLIR Vue Pro and the Zenmuse XT both have a radiometric version that does record absolute temperature. It is recommended to do the processing with the uncompressed Tiff images and create the following index to view absolute temperature.

```
0.04*thermal_ir - 273.15
```

- This also applies (with the same formula) to the newer Wiris camera.

- The Thermomap camera from senseFly also records absolute temperature. The corresponding index is

```
0.01*thermal_ir - 100
```

- This index is already present in the software and is loaded automatically for Thermomap projects.

> **How to get the coefficient of Tiff format? or is the coefficient variable?**

> A **new method** to build the function.

- 1. Use exiftool software (Ubuntu) to get the meta of TIFF or JPG data.

```
exiftool DJI_0777.tif
```

- 2. Find "Central Temperature".

```
exiftool DJI_0777.tif|grep "Central Temperature"
```

- 3. Get the Min/Max Digital Values of TIFF or JPG data from ARCGIS or QGIS.
- 4. Central temperature is the min temperature in my data through the analysis of FLIR Tools, PLEASE NOTICE, this may be different.
- 5. Build a linear equation between Digital Values and Central Temperature.
- 6. Get temperature from TIFF or JPG format data through the equation.
- 7. And then, we can do anything, such as simple operation and modeling using Matlab, R, Python …

[Figure]

Figure 1. The linear equation between Digital Values and Central Temperature.

$$y = 0.0274x - 60.075$$

$$R^2 = 0.95$$

Table 7. UAV flight time during the 2016–2017.

| Flight Date | Flight Time | Height | Slope | Sensor |
|---|---|---|---|---|
| yyyymmdd | hh:mm | m | | |
| 20160417 | 13:36-13:56 | 20-120 | Slopes A and B | RGB |
| 20160830 | 10:18-13:55 | 120 | Slopes A and B | RGB |
| 20170822 | 11:26-13:46 | 120 | Slopes A and B | RGB |
| 20160830 | 12:47-12:52 | 30 | Slope A | TIR |

| Flight Date | Flight Time | Height | Slope | Sensor |
| --- | --- | --- | --- | --- |
| 20170722 | 11:00-15:51 | 150 | Slopes A and B | TIR |
| 20170823 | 10:30-17:25 | 150 | Slopes A and B | TIR |

**Table 8.** Processed UAV data.

| Data Type | Remark |
| --- | --- |
| Boundary | SlopeAB:Shapefile |
| DSM | SM_SlopeAB:Raster |
| Mosaic | Mosaic_SlopeAB:Raster |

**R code of permafrost indices and visualization**

**Script**

**MAAT.R**

- Function for computing Mean Annual Air Temperature (MAAT) index

**MAGST.R**

- Function for computing Mean Annual Ground Surface Temperature (MAGST) index

**Meteorogical.R**

- Plot Meteorogical station observation data, MAAT and MAGST indices

**Data**

The **Data directory** "./Data" contains the following data:

**Table 9.** Data files.

| Data file | Description |
|-----------|-------------|
| Golmud1955-2018.csv | Meteorological observations of Golmud field station |
| Wudaoliang1956-2018.csv | Meteorological observations of Wudaoliang field station |
| XDTMS2014-2018.csv | Meteorological observations of Xidatan field station |
| XDTMS2014-2018_GT.csv | Xidatan field station, ONLY Ground Temperature in different layers |
| XDTMS2014-2018_PREC.csv | Xidatan field station, ONLY Precipitation |
| MAAT_MAGST_Golmud_Wudaoliang_1956-2018.csv | After running MAAT and MAGST, the data of the two field stations need to be merged together for drawing. This data has been manually merged. |

The **output data** is also placed in this directory "./Data".

**Figure**

The output Figures are placed in Figure directory './Figure', and the **operation video** are also placed in this directory.

**Usage**

**Please execute the following statement in Rstudio or R software.**

First, please install **ggplot2** package in Rstudio or R software, and set the environment variables.

```
install.packages('ggplot2')
library('ggplot2')
```

```
**Init**
**clear the environment**
rm(list=ls())
**set workdir**
**setwd('./Script')**
**Data directory**
DataRoot  <- './Data'
**Figure directory**
FigRoot  <- './Figure'
```

> and then run Meteorological.R.

```
source('Meteorological.R')
```

> Or copy the code in Meteorological.R **in turn** and execute it in Rstudio or R software.

> MAAT.R and MAGST.R have been implemented in Meteorological.R, **no additional execution is required.**

```
source('MAAT.R')
source('MAGST.R')
```
* * *
**Operation video**

[Figure]
* * *
**Requirements**

- RStudio Version 1.3.959 or later
- R Statistical Computing Software, 4.0.2 or later
- Package ggplot2 version 3.3.2

**Article DOI**

- https://doi.org/10.5194/essd-2020-106
- This article contains all the data DOI.

**Citation**

Luo, L., Zhuang, Y., Zhang, M., Zhang, Z., Ma, W., Zhao, W., Zhao, L., Wang, L., Shi, Y., Zhang, Z., Duan, Q., Tian, D., and Zhou, Q.: An integrated observation dataset of the hydrological-thermal-deformation dynamics in the permafrost slopes and engineering infrastructure in the Qinghai-Tibet Engineering Corridor, Earth Syst. Sci. Data Discuss. [preprint], https://doi.org/10.5194/essd-2020-106, in review, 2020.

**Abbreviation**

- **TDR:** Time-domain Reflectometer
- **TLS:** Terrestrial Laser Scanning
- **UAV:** Unmanned Aerial Vehicle
- **RGB:** Red-Green-Blue
- **TIR:** Thermal Infrared
- **InSAR:** Interferometric Synthetic Aperture Radar
- **MAAT:** Mean Annual Air Temperature
- **MAGST:** Mean Annual Ground Surface Temperature

**Data resource provider**

**Lihui Luo**

Northwest Institute of Eco-Environment and Resources, Chinese Academy of Sciences

luolh@lzb.ac.cn

**Yanli Zhuang**

Northwest Institute of Eco-Environment and Resources, Chinese Academy of Sciences

zhuangyl@lzb.ac.cn

**Mingyi Zhang**

Northwest Institute of Eco-Environment and Resources, Chinese Academy of Sciences

myzhang@lzb.ac.cn

**Zhongqiong Zhang**

Northwest Institute of Eco-Environment and Resources, Chinese Academy of Sciences

zhangzq@lzb.ac.cn

**Wei Ma**

Northwest Institute of Eco-Environment and Resources, Chinese Academy of Sciences

mawei@lzb.ac.cn

**Wenzhi Zhao**

Northwest Institute of Eco-Environment and Resources, Chinese Academy of Sciences

zhaowzh@lzb.ac.cn

**Lin Zhao**

Northwest Institute of Eco-Environment and Resources, Chinese Academy of Sciences

linzhao@lzb.ac.cn

**Li Wang**

Qinghai Institute of Meteorological Science

liw0209@sohu.com

**Yanmei Shi**

32016 PLA Troops

**Ze Zhang**

Northwest Institute of Eco-Environment and Resources, Chinese Academy of Sciences

zhangze@lzb.ac.cn

**Quntao Duan**

Northwest Institute of Eco-Environment and Resources, Chinese Academy of Sciences

duanqt@lzb.ac.cn

**Deyu Tian**

Northwest Institute of Eco-Environment and Resources, Chinese Academy of Sciences

tiandy@lzb.ac.cn

**Qingguo Zhou**

Lanzhou University

zhouqg@lzu.edu.cn

**Data Sources and Terms of Use**

The use of data is conditional on citing the original data sources. Full details on how to cite the data are given at the bottom of each page. For research projects, if the data are essential to the work, or if an important result or conclusion depends on the data, co-authorship may need to be considered. Permafrost engineering and slope monitoring facilitate the acquisition of data to encourage its use and promote understanding of the potential impact of freeze-thaw cycles on Permafrost engineering. Respecting original data sources is key to help secure the support of data providers to enhance, maintain and update valuable data.

**Acknowledgements**

Funded by the National Natural Science Foundation of China (41871065), the National Science Fund for Distinguished Young Scholars (41825015), the Key Research Project of Frontier Science of Chinese Academy of Sciences (QYZDJ-SSW-DQC040), and the Strategic Priority Research Program of the Chinese Academy of Sciences (XDA19090122).

**License**

Apache License 2.0

**Contact**

**Dr. Lihui Luo**
Northwest Institute of Eco-Environment and Resources, Chinese Academy of Sciences
[luolh@lzb.ac.cn](mailto:luolh@lzb.ac.cn)

updated: 2021/04/14

---

## Author Comment (AC2) · 14 Apr 2021

Response to referee comment 2:

The manuscript by Luo et al. described multiple observation data sets in the Qinghai-Tibet Engineering Corridor (QTEC). I agree with the previous reviewer's comments about the hard-won data in this manuscript. What is particularly commendable is that

the author chose a study area where railway, highway and electrical towers are all distributed on a frozen soil slope. Temperature, air and ground temperature, is the most important indicator of changes in frozen soil. The author uses drones equipped with thermal infrared sensors to monitor spatial changes in surface ground temperature. This data should be relatively rare. This set of data is of great significance for studying the interaction between frozen soil engineering and slopes. Overall, this is a well-prepared manuscript with useful data. The study area is very typical and distinctive.

Response: Thank you for the insightful comments. In revising the paper, we have carefully considered your comments and suggestions. We agree with your comments regarding the metadata, code execution, and data description, among others. To address these concerns, we have made the following modifications to the manuscript: (1) we have added README.md files for the entire dataset of the manuscript and for each data set, such as meteorological and ground observations, TLS measurements, UAV RGB and TIR images, and R code of permafrost indices and visualization, and generated the corresponding README pdf and html files; (2) we have checked the integrity of the data file and added the missing data, including InSAR data and the study area boundary shapefile data in the TLS measurement dataset; (3) we have added vector and raster data of the boundary, DSM (digital surface model), and mosaic of the study area processed by UAV monitoring data; (4) we have renamed some data files because it was difficult for data users to obtain certain data due to naming reasons, and reorganized the file directory, (5) we have modified many inappropriate expressions, including the title; (6) we have updated the data DOI; (7) we have deleted some references with little relevance and added some related references; and (8) we have improved the flow of the language throughout the manuscript (Figure R1). We have tried our best to address each of your points in detail. We feel the revision represents an improvement, and we hope that you agree. For more details, please see our replies below.

Figure R1. Editorial Certificate.

Therefore, I don't have any major suggestions on how to improve the manuscript. Please see some minor comments below.

Minor comments: 1. Please provide a more detailed metadata description of the data set.

Response: We have added metadata files README.md for all datasets and generated the corresponding html and pdf format files. The study area embeds Google Maps in the README.md file. Meteorological and ground observations, as well as the R code of permafrost indices and visualization, include the period from 1955 to 2019. TLS measurements and UAV RGB and TIR images are from 2014 to 2017. We have added a description of the time period in the main text and README.md.

2. It is recommended to add the running notes in the code, and increase the readability of the code, so that users can not only execute, but also modify and improve.

Response: Thank you for the insightful comments. We have reorganized the code, added the required comments and instructions to the code, added a new instruction document on how to use the code, and added the README.md markdown file for operation of the code, including the corresponding html and pdf files. We have also recorded an operation video and provided it in README.md and README.html.

3. Please delete Figure B3. If possible, just describe it in the text.

Response: We have deleted Figure B3.

4. The latest references need to be cited, and some references need to be added. As in the following article: Wu, Q., Sheng, Y., Yu, Q., Chen, J., and Ma, W.: Engineering in the rugged permafrost terrain on the roof of the world under a warming climate, Permafrost and Periglacial Processes, 31, 417-428, https://doi.org/10.1002/ppp.2059, 2020.

Response: We have added the indicated reference and updated some references in the manuscript.

5. This manuscript focuses on ground and drone monitoring data, so it is recommended to delete InSAR data.

Response: As a supplement to the TLS point cloud data, we have prepared Sentinel-1 deformation data for the freeze-thaw stage in the study area from 2014 to 2020 using interferometric synthetic aperture radar (InSAR) technology. These are the InSAR data for the entire study area. These data are a good supplement and comparison to the TLS point cloud data. We still retain these data in the TLS measurement dataset.

References: Bessette-Kirton, E. K., and Coe, J. A.: A 36-Year Record of Rock Avalanches in the Saint Elias Mountains of Alaska, With Implications for Future Hazards, Frontiers in Earth Science, 8, https://doi.org/10.3389/feart.2020.00293, 2020. Guo, D., and Sun, J.: Permafrost Thaw and Associated Settlement Hazard Onset Timing over the Qinghai-Tibet Engineering Corridor, International Journal of Disaster Risk Science, 6, 347-358, https://doi.org/10.1007/s13753-015-0072-3, 2015. Huggel, C., Salzmann, N., Allen, S., Caplan-Auerbach, J., Fischer, L., Haeberli, W., Larsen, C., Schneider, D., and Wessels, R.: Recent and future warm extreme events and high-mountain slope stability, Philosophical Transactions of the Royal Society A: Mathematical, Physical and Engineering Sciences, 368, 2435-2459, https://doi.org/10.1098/rsta.2010.0078, 2010. Liu, G., Xie, C., Zhao, L., Xiao, Y., Wu, T., Wang, W., and Liu, W.: Permafrost warming near the northern limit of permafrost on the Qinghai–Tibetan Plateau during the period from 2005 to 2017: A case study in the Xidatan area, Permafrost and Periglacial Processes, https://doi.org/10.1002/ppp.2089, 2020. Luo, L., Ma, W., Zhao, W., Zhuang, Y., Zhang, Z., Zhang, M., Ma, D., and Zhou, Q.: UAV-based spatiotemporal thermal patterns of permafrost slopes along the Qinghai–Tibet Engineering Corridor, Landslides, 15, 2161–2172, https://doi.org/10.1007/s10346-018-1028-7, 2018a. Luo, L., Zhang, Z., Ma, W., Yi, S., and Zhuang, Y.: PIC v1.3: comprehensive R package for computing permafrost indices with daily weather observations and atmospheric forcing over the Qinghai–Tibet Plateau, Geosci Model Dev, 11, 2475-2491, https://doi.org/10.5194/gmd-11-2475-

2018, 2018b. Ma, W., Niu, F., Akagawa, S., and Jin, D.: Slope instability phenomena in permafrost regions of Qinghai-Tibet Plateau, China, Landslides, 3, 260-264, https://doi.org/10.1007/s10346-006-0045-0, 2006. Ma, W., Mu, Y., Zhang, J., Yu, W., Zhou, Z., and Chen, T.: Lateral thermal influences of roadway and railway embankments in permafrost zones along the Qinghai-Tibet Engineering Corridor, Transportation Geotechnics, 21, https://doi.org/10.1016/j.trgeo.2019.100285, 2019. Niu, F., Luo, J., Lin, Z., Fang, J., and Liu, M.: Thaw-induced slope failures and stability analyses in permafrost regions of the Qinghai-Tibet Plateau, China, Landslides, 13, 55-65, https://doi.org/10.1007/s10346-014-0545-2, 2015. Obu, J., Westermann, S., Bartsch, A., Berdnikov, N., Christiansen, H. H., Dashtseren, A., Delaloye, R., Elberling, B., Etzelmüller, B., Kholodov, A., Khomutov, A., Kääb, A., Leibman, M. O., Lewkowicz, A. G., Panda, S. K., Romanovsky, V., Way, R. G., Westergaard-Nielsen, A., Wu, T., Yamkhin, J., and Zou, D.: Northern Hemisphere permafrost map based on TTOP modelling for 2000–2016 at 1 km2 scale, Earth-Science Reviews, 193, 299-316, https://doi.org/10.1016/j.earscirev.2019.04.023, 2019. Patton, A. I., Rathburn, S. L., and Capps, D. M.: Landslide response to climate change in permafrost regions, Geomorphology, 340, 116-128, https://doi.org/10.1016/j.geomorph.2019.04.029, 2019. Qiu, G., Zhou, Y., Guo, D., and Wang, Y.: The map of geocryological regionalization and classification in China, Science Press, Beijing (in Chinese), 2000. Spaans, E. J. A., and Baker, J. M.: Examining the use of time domain reflectometry for measuring liquid water content in frozen soil, Water Resour Res, 31, 2917-2925, https://doi.org/10.1029/95wr02769, 1995. Streletskiy, D. A., Suter, L. J., Shiklomanov, N. I., Porfiriev, B. N., and Eliseev, D. O.: Assessment of climate change impacts on buildings, structures and infrastructure in the Russian regions on permafrost, Environ Res Lett, 14, https://doi.org/10.1088/1748-9326/aaf5e6, 2019. Wang, S., Niu, F., Chen, J., and Dong, Y.: Permafrost research in China related to express highway construction, Permafrost and Periglacial Processes, 31, 406-416, https://doi.org/10.1002/ppp.2053, 2020. Wirz, V., Geertsema, M., Gruber, S., and Purves, R. S.: Temporal variability of diverse mountain permafrost slope movements derived from multi-year daily GPS

data, Mattertal, Switzerland, Landslides, 13, 67-83, https://doi.org/10.1007/s10346-014-0544-3, 2015. Wu, Q., Dong, X., Liu, Y., and Jin, H.: Responses of Permafrost on the Qinghai-Tibet Plateau, China, to Climate Change and Engineering Construction, Arctic, Antarctic, and Alpine Research, 39, 682-687, https://doi.org/10.1657/1523-0430(07-508)[wu]2.0.Co;2, 2007. Wu, Q., and Zhang, T.: Recent permafrost warming on the Qinghai-Tibetan Plateau, Journal of Geophysical Research, 113, https://doi.org/10.1029/2007jd009539, 2008. Yang, Y.-z., Wu, Q.-b., Deng, Y.-s., Jiang, G.-l., and Zhang, P.: Chemical Composition of Borehole Gas in Kunlun Pass Basin in Permafrost Regions in Qinghai-Tibet Plateau, Natural Gas Geoscience, 6, 2011. Yang, Y., Wu, Q., Jiang, G., and Zhang, P.: Stable Isotopic Stratification and Growth Patterns of Ground Ice in Permafrost on the Qinghai-Tibet Plateau, China, Permafrost and Periglacial Processes, 28, 119-129, https://doi.org/10.1002/ppp.1892, 2017. Yu, W., Zhang, T., Lu, Y., Han, F., Zhou, Y., and Hu, D.: Engineering risk analysis in cold regions: State of the art and perspectives, Cold Regions Science and Technology, 171, https://doi.org/10.1016/j.coldregions.2019.102963, 2020. Yuan, C., Yu, Q., You, Y., and Guo, L.: Deformation mechanism of an expressway embankment in warm and high ice content permafrost regions, Appl Therm Eng, 121, 1032-1039, https://doi.org/10.1016/j.applthermaleng.2017.04.128, 2017. Zhang, Z., Yu, Q., You, Y., Guo, L., Wang, X., Liu, G., and Wu, G.: Cooling effect analysis of temperature-controlled ventilated embankment in Qinghai-Tibet testing expressway, Cold Regions Science and Technology, 173, https://doi.org/10.1016/j.coldregions.2020.103012, 2020. Zhao, L., Zou, D., Hu, G., Du, E., Pang, Q., Xiao, Y., Li, R., Sheng, Y., Wu, X., Sun, Z., Wang, L., Wang, C., Ma, L., Zhou, H., and Liu, S.: Changing climate and the permafrost environment on the Qinghai–Tibet (Xizang) plateau, Permafrost and Periglacial Processes, 31, 396-405, https://doi.org/10.1002/ppp.2056, 2020.

Please also note the supplement to this comment:
https://essd.copernicus.org/preprints/essd-2020-106/essd-2020-106-AC2-supplement.pdf

**Supplement:**

**Authors reply to the comments by Anonymous Referee of the manuscript essd-2020-106**

**"An integrated observation dataset of the hydrological-thermal-deformation dynamics in the permafrost slopes and engineering infrastructure in the Qinghai-Tibet Engineering Corridor"**

**by Lihui Luo et al.**

We thank Anonymous Referee #2 for valuable feedback, which helped us improve our manuscript. Please find below the Reviewer comments in black, Author responses in green, and Changes to the manuscript in blue.

**Response to referee comment 2:**

The manuscript by Luo et al. described multiple observation data sets in the Qinghai-Tibet Engineering Corridor (QTEC). I agree with the previous reviewer's comments about the hard-won data in this manuscript. What is particularly commendable is that the author chose a study area where railway, highway and electrical towers are all distributed on a frozen soil slope. Temperature, air and ground temperature, is the most important indicator of changes in frozen soil. The author uses drones equipped with thermal infrared sensors to monitor spatial changes in surface ground temperature. This data should be relatively rare. This set of data is of great significance for studying the interaction between frozen soil engineering and slopes. Overall, this is a well-prepared manuscript with useful data. The study area is very typical and distinctive.

Thank you for the insightful comments. In revising the paper, we have carefully considered your comments and suggestions. We agree with your comments regarding the metadata, code execution, and data description, among others. To address these concerns, we have made the following modifications to

the manuscript: (1) we have added README.md files for the entire dataset of the manuscript and for each data set, such as meteorological and ground observations, TLS measurements, UAV RGB and TIR images, and R code of permafrost indices and visualization, and generated the corresponding README pdf and html files; (2) we have checked the integrity of the data file and added the missing data, including InSAR data and the study area boundary shapefile data in the TLS measurement dataset; (3) we have added vector and raster data of the boundary, DSM (digital surface model), and mosaic of the study area processed by UAV monitoring data; (4) we have renamed some data files because it was difficult for data users to obtain certain data due to naming reasons, and reorganized the file directory, (5) we have modified many inappropriate expressions, including the title; (6) we have updated the data DOI; (7) we have deleted some references with little relevance and added some related references; and (8) we have improved the flow of the language throughout the manuscript (Figure R1). We have tried our best to address each of your points in detail. We feel the revision represents an improvement, and we hope that you agree. For more details, please see our replies below.

[Figure]

Figure R1. Editorial Certificate.

Therefore, I don't have any major suggestions on how to improve the manuscript. Please see some minor

comments below.

Minor comments:

1. Please provide a more detailed metadata description of the data set.

We have added metadata files README.md for all datasets and generated the corresponding html and pdf format files. The study area embeds Google Maps in the README.md file. Meteorological and ground observations, as well as the R code of permafrost indices and visualization, include the period from 1955 to 2019. TLS measurements and UAV RGB and TIR images are from 2014 to 2017. We have added a description of the time period in the main text and README.md.

2. It is recommended to add the running notes in the code, and increase the readability of the code, so that users can not only execute, but also modify and improve.

Thank you for the insightful comments. We have reorganized the code, added the required comments and instructions to the code, added a new instruction document on how to use the code, and added the README.md markdown file for operation of the code, including the corresponding html and pdf files. We have also recorded an operation video and provided it in README.md and README.html.

3. Please delete Figure B3. If possible, just describe it in the text.

We have deleted Figure B3.

4. The latest references need to be cited, and some references need to be added. As in the following article:

Wu, Q., Sheng, Y., Yu, Q., Chen, J., and Ma, W.: Engineering in the rugged permafrost terrain on the roof of the world under a warming climate, Permafrost and Periglacial Processes, 31, 417-428, https://doi.org/10.1002/ppp.2059, 2020.

We have added the indicated reference and updated some references in the manuscript.

5. This manuscript focuses on ground and drone monitoring data, so it is recommended to delete InSAR data.

As a supplement to the TLS point cloud data, we have prepared Sentinel-1 deformation data for the

freeze-thaw stage in the study area from 2014 to 2020 using interferometric synthetic aperture radar (InSAR) technology. These are the InSAR data for the entire study area. These data are a good supplement and comparison to the TLS point cloud data. We still retain these data in the TLS measurement dataset.

**References:**

Bessette-Kirton, E. K., and Coe, J. A.: A 36-Year Record of Rock Avalanches in the Saint Elias Mountains of Alaska, With Implications for Future Hazards, Frontiers in Earth Science, 8, https://doi.org/10.3389/feart.2020.00293, 2020.

Guo, D., and Sun, J.: Permafrost Thaw and Associated Settlement Hazard Onset Timing over the Qinghai-Tibet Engineering Corridor, International Journal of Disaster Risk Science, 6, 347-358, https://doi.org/10.1007/s13753-015-0072-3, 2015.

Huggel, C., Salzmann, N., Allen, S., Caplan-Auerbach, J., Fischer, L., Haeberli, W., Larsen, C., Schneider, D., and Wessels, R.: Recent and future warm extreme events and high-mountain slope stability, Philosophical Transactions of the Royal Society A: Mathematical, Physical and Engineering Sciences, 368, 2435-2459, https://doi.org/10.1098/rsta.2010.0078, 2010.

Liu, G., Xie, C., Zhao, L., Xiao, Y., Wu, T., Wang, W., and Liu, W.: Permafrost warming near the northern limit of permafrost on the Qinghai–Tibetan Plateau during the period from 2005 to 2017: A case study in the Xidatan area, Permafrost and Periglacial Processes, https://doi.org/10.1002/ppp.2089, 2020.

Luo, L., Ma, W., Zhao, W., Zhuang, Y., Zhang, Z., Zhang, M., Ma, D., and Zhou, Q.: UAV-based spatiotemporal thermal patterns of permafrost slopes along the Qinghai–Tibet Engineering Corridor, Landslides, 15, 2161–2172, https://doi.org/10.1007/s10346-018-1028-7, 2018a.

Luo, L., Zhang, Z., Ma, W., Yi, S., and Zhuang, Y.: PIC v1.3: comprehensive R package for computing permafrost indices with daily weather observations and atmospheric forcing over the Qinghai–Tibet Plateau, Geosci Model Dev, 11, 2475-2491, https://doi.org/10.5194/gmd-11-2475-2018, 2018b.

Ma, W., Niu, F., Akagawa, S., and Jin, D.: Slope instability phenomena in permafrost regions of Qinghai-Tibet Plateau, China, Landslides, 3, 260-264, https://doi.org/10.1007/s10346-006-0045-0, 2006.

Ma, W., Mu, Y., Zhang, J., Yu, W., Zhou, Z., and Chen, T.: Lateral thermal influences of roadway and railway embankments in permafrost zones along the Qinghai-Tibet Engineering Corridor,

Transportation Geotechnics, 21, https://doi.org/10.1016/j.trgeo.2019.100285, 2019.

Niu, F., Luo, J., Lin, Z., Fang, J., and Liu, M.: Thaw-induced slope failures and stability analyses in permafrost regions of the Qinghai-Tibet Plateau, China, Landslides, 13, 55-65, https://doi.org/10.1007/s10346-014-0545-2, 2015.

Obu, J., Westermann, S., Bartsch, A., Berdnikov, N., Christiansen, H. H., Dashtseren, A., Delaloye, R., Elberling, B., Etzelmüller, B., Kholodov, A., Khomutov, A., Kääb, A., Leibman, M. O., Lewkowicz, A. G., Panda, S. K., Romanovsky, V., Way, R. G., Westergaard-Nielsen, A., Wu, T., Yamkhin, J., and Zou, D.: Northern Hemisphere permafrost map based on TTOP modelling for 2000–2016 at 1 km2 scale, Earth-Science Reviews, 193, 299-316, https://doi.org/10.1016/j.earscirev.2019.04.023, 2019.

Patton, A. I., Rathburn, S. L., and Capps, D. M.: Landslide response to climate change in permafrost regions, Geomorphology, 340, 116-128, https://doi.org/10.1016/j.geomorph.2019.04.029, 2019.

Qiu, G., Zhou, Y., Guo, D., and Wang, Y.: The map of geocryological regionalization and classification in China, Science Press, Beijing (in Chinese), 2000.

Spaans, E. J. A., and Baker, J. M.: Examining the use of time domain reflectometry for measuring liquid water content in frozen soil, Water Resour Res, 31, 2917-2925, https://doi.org/10.1029/95wr02769, 1995.

Streletskiy, D. A., Suter, L. J., Shiklomanov, N. I., Porfiriev, B. N., and Eliseev, D. O.: Assessment of climate change impacts on buildings, structures and infrastructure in the Russian regions on permafrost, Environ Res Lett, 14, https://doi.org/10.1088/1748-9326/aaf5e6, 2019.

Wang, S., Niu, F., Chen, J., and Dong, Y.: Permafrost research in China related to express highway construction, Permafrost and Periglacial Processes, 31, 406-416, https://doi.org/10.1002/ppp.2053, 2020.

Wirz, V., Geertsema, M., Gruber, S., and Purves, R. S.: Temporal variability of diverse mountain permafrost slope movements derived from multi-year daily GPS data, Mattertal, Switzerland, Landslides, 13, 67-83, https://doi.org/10.1007/s10346-014-0544-3, 2015.

Wu, Q., Dong, X., Liu, Y., and Jin, H.: Responses of Permafrost on the Qinghai-Tibet Plateau, China, to Climate Change and Engineering Construction, Arctic, Antarctic, and Alpine Research, 39, 682-687, https://doi.org/10.1657/1523-0430(07-508)[wu]2.0.Co;2, 2007.

Wu, Q., and Zhang, T.: Recent permafrost warming on the Qinghai-Tibetan Plateau, Journal of

Geophysical Research, 113, https://doi.org/10.1029/2007jd009539, 2008.

Yang, Y.-z., Wu, Q.-b., Deng, Y.-s., Jiang, G.-l., and Zhang, P.: Chemical Composition of Borehole Gas in Kunlun Pass Basin in Permafrost Regions in Qinghai-Tibet Plateau, Natural Gas Geoscience, 6, 2011.

Yang, Y., Wu, Q., Jiang, G., and Zhang, P.: Stable Isotopic Stratification and Growth Patterns of Ground Ice in Permafrost on the Qinghai-Tibet Plateau, China, Permafrost and Periglacial Processes, 28, 119-129, https://doi.org/10.1002/ppp.1892, 2017.

Yu, W., Zhang, T., Lu, Y., Han, F., Zhou, Y., and Hu, D.: Engineering risk analysis in cold regions: State of the art and perspectives, Cold Regions Science and Technology, 171, https://doi.org/10.1016/j.coldregions.2019.102963, 2020.

Yuan, C., Yu, Q., You, Y., and Guo, L.: Deformation mechanism of an expressway embankment in warm and high ice content permafrost regions, Appl Therm Eng, 121, 1032-1039, https://doi.org/10.1016/j.applthermaleng.2017.04.128, 2017.

Zhang, Z., Yu, Q., You, Y., Guo, L., Wang, X., Liu, G., and Wu, G.: Cooling effect analysis of temperature-controlled ventilated embankment in Qinghai-Tibet testing expressway, Cold Regions Science and Technology, 173, https://doi.org/10.1016/j.coldregions.2020.103012, 2020.

Zhao, L., Zou, D., Hu, G., Du, E., Pang, Q., Xiao, Y., Li, R., Sheng, Y., Wu, X., Sun, Z., Wang, L., Wang, C., Ma, L., Zhou, H., and Liu, S.: Changing climate and the permafrost environment on the Qinghai–Tibet (Xizang) plateau, Permafrost and Periglacial Processes, 31, 396-405, https://doi.org/10.1002/ppp.2056, 2020.

**Data description for essd-2020-106**

**An integrated observation dataset of the hydrological-thermal-deformation in the permafrost slopes and engineering infrastructure in the Qinghai-Tibet Engineering Corridor**

**Description**

> **Meteorological observations** Observation of meteorological factors was conducted at two permanent meteorological stations (Golmud and Wudaoliang) and one field meteorological station (Xidatan) with daily meteorological records. All three meteorological stations contain ground observations.

> **Ground observations** The ground temperature and moisture data from the near-surface to within 270 cm in the active layer were recorded. In situ ground observations were deployed starting in July 2013 using thermocouple probes (105T, Campbell Scientific) to measure the soil temperature and using 11 time-domain reflectometer (TDR) probes (model CS615-L, Campbell Scientific) to measure the soil volumetric water content.

> **TLS measurements** A FARO Focus3D X130 3D laser scanner and six Trimble 5700 GNSS systems were deployed around permafrost slopes between May 2014 and October 2015. As a supplement to the TLS point cloud data, we used Interferometric Synthetic Aperture Radar (InSAR) technology to prepare Sentinel-1 deformation data for the study area from 2014 to 2020.

> **UAV RGB and TIR images** Two permafrost slopes were conducted four flight experiments with UAV-mounted RGB and TIR sensors in 2016 and 2017.

> **R code of permafrost indices and visualization** R Script for plotting meteorological observation data and permafrost indices (MAAT and MAGST) during 1955-2018.

**Keywords**

**Theme:** Permafrost slope; Permafrost engineering; Freeze-thaw; hydrological-thermal-Deformation; Qinghai-Tibet plateau

**Discipline:** cryosphere; In-situ monitoring data; Remote sensing data using TLS and UAV

**Places:** Qinghai-Tibet Engineering Corridor; Kunlun Mountain Pass close to Hoh Xil Nature Reserve

**Data details**

**Scale:** UAV RGB: ~5 cm; UAV TIR: ~ 20 cm; TLS measurements: 0.009°

**Coordinate Reference System:** EPSG: 4326 - WGS 84

**Filesize:**~ 5 G

**Data format:** GeoTiff, CSV, EXCEL XLSX, TXT, WRP, Tif, JPG

**Space scope**

```
                North: 35°39′ 10″
    West: 90°3′ 30″     –      East: 90°3′ 55″
                South: 35°38′ 35″
```

[Figure]

**Time period**

Table 1. Observations period of all datasets.

| Data Type | Location | Period | Remark |
|---|---|---|---|
| Meteorological observations | Golmud station | 1955-2018 | National Reference Station |
| Meteorological observations | Xidatan station | 2014-2018 | National General Station |
| Meteorological observations | Wudaoliang station | 1956-2018 | National Reference Station |
| Ground observations | Study Area | 2014-2019 | Field test site |
| Ground observations | Golmud station | 1955-2018 | National Reference Station |

| Data Type | Location | Period | Remark |
|---|---|---|---|
| Ground observations | Xidatan station | 2014-2018 | National General Station |
| Ground observations | Wudaoliang station | 1956-2018 | National Reference Station |
| TLS measurements | Study Area | 2014-2015 | Contains measurement and comparative analysis data |
| InSAR | Study Area | 2014-2020 | Contains thawing and freezing period data |
| UAV RGB and TIR images | Study Area | 2016-2017 | tif & jpg can be processed by Pix4Dmapper & FLIR |
| R code of permafrost indices and visualization | Stations | 1955-2018 | Plot Fig. 2 & F1; Computing MAAT & MAGST |

**Meteorological and Ground observations**

**Table 2.** Observations period of datasets.

| Data Type | Location | Period | File Names |
|---|---|---|---|
| Meteorological observations | Golmud station | 1955-2018 | Meteo_52818_Golmud_1955-2010.dat;Meteo_52818_Golmud_2010-2018.xlsx |
| Meteorological observations | Xidatan station | 2014-2018 | Meteo_00000_Golmud_2014-2019.xlsx |
| Meteorological observations | Wudaoliang station | 1956-2018 | Meteo_52908_Wudaoliang_1956-2010.dat;Meteo_52908_Wudaoliang_2010-2018.xlsx |
| Ground observations | Study Area | 2014-2019 | GT00000_Slopes_2014-2019.xlsx |

| Data Type | Location | Period | File Names |
|-----------|----------|--------|------------|
| Ground observations | Golmud station | 1955-2018 | GT52818_Golmud.txt |
| Ground observations | Xidatan station | 2014-2018 | Meteo_00000_Xidatan_2014-2019.xlsx |
| Ground observations | Wudaoliang station | 1956-2018 | GT52908_Wudaoliang.txt |

**Table 3.** Ground data Metadata of meteorological stations data. The file name with **'GT'** is ground observation data.

| | ID | Variable | Type | Field Name | Unit | Description |
|---|----|----------|------|------------|------|-------------|
| 1 | 1 | Station ID | Number(5) | V01000 | | |
| 2 | 5 | Year | Number(4) | V04001 | Year | |
| 3 | 6 | Month | Number(2) | V04002 | Month | |
| 4 | 7 | Day | Number(2) | V04003 | Day | |
| 5 | 32 | Evaporation | Number(6) | V13241 | 0.1mm | evaporation |
| 6 | 53 | average ground temperature at 0 cm | Number(6) | V12240 | 0.1℃ | GT_0_AVG |
| 7 | 54 | daily maximum ground temperature at 0 cm | Number(6) | V12213 | 0.1℃ | GT_0_MAX |
| 8 | 56 | daily minimum ground temperature at 0 cm | Number(6) | V12214 | 0.1℃ | GT_0_MIN |

| | ID | Variable | Type | Field Name | Unit | Description |
|---|---|---|---|---|---|---|
| 9 | 58 | average ground temperature at 5 cm | Number(6) | V12240_005 | 0.1℃ | GT_5_AVG |
| 10 | 59 | average ground temperature at 10 cm | Number(6) | V12240_010 | 0.1℃ | GT_10_AVG |
| 11 | 60 | average ground temperature at 15 cm | Number(6) | V12240_015 | 0.1℃ | GT_15_AVG |
| 12 | 61 | average ground temperature at 20 cm | Number(6) | V12240_020 | 0.1℃ | GT_20_AVG |
| 13 | 62 | average ground temperature at 40 cm | Number(6) | V12240_040 | 0.1℃ | GT_40_AVG |
| 14 | 63 | average ground temperature at 50 cm | Number(6) | V12240_050 | 0.1℃ | GT_50_AVG |
| 15 | 64 | average ground temperature at 80 cm | Number(6) | V12240_080 | 0.1℃ | GT_80_AVG |
| 16 | 65 | average ground temperature at 160 cm | Number(6) | V12240_160 | 0.1℃ | GT_160_AVG |
| 17 | 66 | average ground temperature at 320 cm | Number(6) | V12240_320 | 0.1℃ | GT_320_AVG |

| | ID | Variable | Type | Field Name | Unit | Description |
|---|----|----------|------|------------|------|-------------|
| | | | | | | |

**Table 4.** Meteorological Metadata of meteorological stations data. The file name with **'Meteo'** is Meteorological observation data.

| | ID | Variable | Type | Unit | Description |
|---|----|----------|------|------|-------------|
| 1 | 1 | Station ID | Number(5) | | |
| 2 | 5 | Year | Number(4) | Year | Year |
| 3 | 6 | Month | Number(2) | Month | Mon |
| 4 | 7 | Day | Number(2) | Day | Day |
| 5 | 32 | daily mean air temperature at 2 m | Number(6) | 0.1℃ | Temperate |
| 6 | 53 | maximum air temperature at 2 m | Number(6) | 0.1℃ | Tmax |
| 7 | 54 | minimum air temperature at 2 m | Number(6) | 0.1℃ | Tmin |
| 8 | 56 | average wind speed | Number(6) | 0.1℃ | Wind |
| 9 | 58 | average precipitation | Number(6) | 0.1mm | Precip |
| 10 | 59 | Corrected average precipitation | Number(6) | 0.1℃ | Corrected_P |
| 11 | 60 | Evaporation | Number(6) | 0.1mm | Evaporation |
| 12 | 61 | Air humidity | Number(6) | % | Humidity |
| 13 | 62 | Air pressure | Number(6) | 0.1Pa | Press |
| 14 | 63 | sunshine time | Number(6) | 0.1h | Sunshine |
| 15 | 64 | average ground temperature at 0 cm | Number(6) | 0.1℃ | GT |

**TLS measurements**

**TLS measurements** There are a total of 4 monitorings between May 2014 and October 2015 within two thawing periods and a freezing period. The three freeze-thaw phases are referred to as "first thawing" (May 2014 to October 2014, called here "period 2-1"), "first

freezing" (October 2014 to May 2015, called here "period 3-2"), "second thawing" (May 2015 to October 2015, called here "period 4-3"), "one thawing and one freezing stage" (May 2014 to May 2015, called here "period 3-1"), and "two thawing and one freezing stage" (May 2014 to October 2015, called here "period 4-1") in the following. The file directories for each monitoring are: first, second, third, and fourth. And the file also contains comparative analysis data of different periods.

**Table 5** Freeze-thaw stages of TLS scanner data.

| Status | Condition | Date Span | Days | Slope | Data points |
|--------|-----------|-----------|------|-------|-------------|
| Period 2-1 | Thawing | 05/02/2014–10/10/2014 | 161 | Slope A | 1251706 |
| Period 2-1 | Thawing | 05/02/2014–10/10/2014 | 161 | Slope B | 1367438 |
| Period 3-2 | Freezing | 10/10/2014–05/03/2015 | 205 | Slope A | 1291356 |
| Period 3-2 | Freezing | 10/10/2014–05/03/2015 | 205 | Slope B | 1366141 |
| Period 4-3 | Thawing | 05/03/2015–10/04/2015 | 154 | Slope A | 1248325 |
| Period 4-3 | Thawing | 05/03/2015–10/04/2015 | 154 | Slope B | 1382768 |
| Period 3-1 | one thawing and one freezing | 05/02/2014–05/03/2015 | 366 | Slope A | 1278448 |
| Period 3-1 | one thawing and one freezing | 05/02/2014–05/03/2015 | 366 | Slope B | 1279204 |
| Period 4-1 | two thawing and one freezing | 05/02/2014–10/04/2015 | 520 | Slope A | 1279706 |
| Period 4-1 | two thawing and one freezing | 05/02/2014–10/04/2015 | 520 | Slope B | 1207493 |

**InSAR data** The Sentinel-1 mission provides data from a dual-polarization C-band Synthetic Aperture Radar (SAR) instrument. This collection includes the S1 Ground Range Detected (GRD) scenes, processed using the Sentinel-1 Toolbox to generate a calibrated, ortho-corrected product. File directory is InSAR.

**Table 6.** InSAR data for Permafrost slope A & B, including the study area vector shapefile file(SlopeAB). Direction of the orbit ('ASCENDING' or 'DESCENDING') for the oldest image data in the product (the start of the product). The spatial resolution is 10 meters.

| Data Type | Period | Condition | Remark |
|---|---|---|---|
| asc | 2014-2016 | Tawing | ASCENDING |
| asc | 2014-2017 | Freezing | ASCENDING |
| asc | 2017-2019 | Tawing | ASCENDING |
| asc | 2017-2020 | Freezing | ASCENDING |
| desc | 2014-2016 | Tawing | DESCENDING |
| desc | 2014-2017 | Freezing | DESCENDING |
| desc | 2017-2019 | Tawing | DESCENDING |
| desc | 2017-2020 | Freezing | DESCENDING |
| Study Area boundary | | | SlopeAB:Shapefile |

**UAV RGB and TIR images**

For these two slopes, we conducted four flight experiments with UAV-mounted RGB and TIR sensors. The directory of flight images for RGB and thermal infrared sensors is RGB and TIR.

There are three directories under the RGB directory: **20160417, 20160830 and 20170822**, the format is yyyyymmdd, which represent the UAV photos taken by the RGB camera that day. Please use **exiftool** to view the metadata information of pictures such as timestamp and location.

There are three directories under the TIR directory: **2016SlopeA and 2017SlopeAB**, the format is yyyyySlope, which represent UAV photos taken by the TIR sensor of the year.

> Please use exiftool to view the metadata information of pictures such as timestamp, location, and center point temperature.

> To obtain temperatures, a sensor that is able to provide absolute temperature is needed (instead of relative temperature). The FLIR Vue Pro and the Zenmuse XT do not provide absolute temperature. However, the FLIR Vue Pro and the Zenmuse XT both have a radiometric version that does record absolute temperature. It is recommended to do the processing with the uncompressed Tiff images and create the following index to view absolute temperature.

```
0.04*thermal_ir - 273.15
```

- This also applies (with the same formula) to the newer Wiris camera.

- The Thermomap camera from senseFly also records absolute temperature. The corresponding index is

```
0.01*thermal_ir - 100
```

- This index is already present in the software and is loaded automatically for Thermomap projects.

> **How to get the coefficient of Tiff format? or is the coefficient variable?**

> A **new method** to build the function.

- 1. Use exiftool software (Ubuntu) to get the meta of TIFF or JPG data.

```
exiftool DJI_0777.tif
```

- 2. Find "Central Temperature".

```
exiftool DJI_0777.tif|grep "Central Temperature"
```

- 3. Get the Min/Max Digital Values of TIFF or JPG data from ARCGIS or QGIS.
- 4. Central temperature is the min temperature in my data through the analysis of FLIR Tools, PLEASE NOTICE, this may be different.
- 5. Build a linear equation between Digital Values and Central Temperature.
- 6. Get temperature from TIFF or JPG format data through the equation.
- 7. And then, we can do anything, such as simple operation and modeling using Matlab, R, Python …

[Figure]

Figure 1. The linear equation between Digital Values and Central Temperature.

Table 7. UAV flight time during the 2016–2017.

| Flight Date | Flight Time | Height | Slope | Sensor |
|---|---|---|---|---|
| yyyymmdd | hh:mm | m | | |
| 20160417 | 13:36-13:56 | 20-120 | Slopes A and B | RGB |
| 20160830 | 10:18-13:55 | 120 | Slopes A and B | RGB |
| 20170822 | 11:26-13:46 | 120 | Slopes A and B | RGB |
| 20160830 | 12:47-12:52 | 30 | Slope A | TIR |

| Flight Date | Flight Time | Height | Slope | Sensor |
|---|---|---|---|---|
| 20170722 | 11:00-15:51 | 150 | Slopes A and B | TIR |
| 20170823 | 10:30-17:25 | 150 | Slopes A and B | TIR |

**Table 8.** Processed UAV data.

| Data Type | Remark |
|---|---|
| Boundary | SlopeAB:Shapefile |
| DSM | SM_SlopeAB:Raster |
| Mosaic | Mosaic_SlopeAB:Raster |

**R code of permafrost indices and visualization**

**Script**

**MAAT.R**

- Function for computing Mean Annual Air Temperature (MAAT) index

**MAGST.R**

- Function for computing Mean Annual Ground Surface Temperature (MAGST) index

**Meteorogical.R**

- Plot Meteorogical station observation data, MAAT and MAGST indices

**Data**

The **Data directory** "./Data" contains the following data:

> **Table 9.** Data files.

| Data file | Description |
| --- | --- |
| Golmud1955-2018.csv | Meteorological observations of Golmud field station |
| Wudaoliang1956-2018.csv | Meteorological observations of Wudaoliang field station |
| XDTMS2014-2018.csv | Meteorological observations of Xidatan field station |
| XDTMS2014-2018_GT.csv | Xidatan field station, ONLY Ground Temperature in different layers |
| XDTMS2014-2018_PREC.csv | Xidatan field station, ONLY Precipitation |
| MAAT_MAGST_Golmud_Wudaoliang_1956-2018.csv | After running MAAT and MAGST, the data of the two field stations need to be merged together for drawing. This data has been manually merged. |

> The **output data** is also placed in this directory "./Data".

**Figure**

> The output Figures are placed in Figure directory './Figure', and the **operation video** are also placed in this directory.

**Usage**

> **Please execute the following statement in Rstudio or R software.**

> First, please install **ggplot2** package in Rstudio or R software, and set the environment variables.

```
install.packages('ggplot2')
library('ggplot2')
```

```
**Init**
**clear the environment**
rm(list=ls())
**set workdir**
**setwd('./Script')**
**Data directory**
DataRoot  <- './Data'
**Figure directory**
FigRoot  <- './Figure'
```

and then run Meteorological.R.

```
source('Meteorological.R')
```

Or copy the code in Meteorological.R **in turn** and execute it in Rstudio or R software.

MAAT.R and MAGST.R have been implemented in Meteorological.R, **no additional execution is required.**

```
source('MAAT.R')
source('MAGST.R')
```

**Operation video**

[Figure]

**Requirements**

- RStudio Version 1.3.959 or later
- R Statistical Computing Software, 4.0.2 or later
- Package ggplot2 version 3.3.2

**Article DOI**

- https://doi.org/10.5194/essd-2020-106
- This article contains all the data DOI.

**Citation**

Luo, L., Zhuang, Y., Zhang, M., Zhang, Z., Ma, W., Zhao, W., Zhao, L., Wang, L., Shi, Y., Zhang, Z., Duan, Q., Tian, D., and Zhou, Q.: An integrated observation dataset of the hydrological-thermal-deformation dynamics in the permafrost slopes and engineering infrastructure in the Qinghai-Tibet Engineering Corridor, Earth Syst. Sci. Data Discuss. [preprint], https://doi.org/10.5194/essd-2020-106, in review, 2020.

**Abbreviation**

- **TDR:** Time-domain Reflectometer
- **TLS:** Terrestrial Laser Scanning
- **UAV:** Unmanned Aerial Vehicle
- **RGB:** Red-Green-Blue
- **TIR:** Thermal Infrared
- **InSAR:** Interferometric Synthetic Aperture Radar
- **MAAT:** Mean Annual Air Temperature
- **MAGST:** Mean Annual Ground Surface Temperature

**Data resource provider**

**Lihui Luo**

Northwest Institute of Eco-Environment and Resources, Chinese Academy of Sciences

luolh@lzb.ac.cn

**Yanli Zhuang**

Northwest Institute of Eco-Environment and Resources, Chinese Academy of Sciences

zhuangyl@lzb.ac.cn

**Mingyi Zhang**

Northwest Institute of Eco-Environment and Resources, Chinese Academy of Sciences

myzhang@lzb.ac.cn

**Zhongqiong Zhang**

Northwest Institute of Eco-Environment and Resources, Chinese Academy of Sciences

zhangzq@lzb.ac.cn

**Wei Ma**

Northwest Institute of Eco-Environment and Resources, Chinese Academy of Sciences

mawei@lzb.ac.cn

**Wenzhi Zhao**

Northwest Institute of Eco-Environment and Resources, Chinese Academy of Sciences

zhaowzh@lzb.ac.cn

**Lin Zhao**

Northwest Institute of Eco-Environment and Resources, Chinese Academy of Sciences

linzhao@lzb.ac.cn

**Li Wang**

Qinghai Institute of Meteorological Science

liw0209@sohu.com

**Yanmei Shi**

32016 PLA Troops

**Ze Zhang**

Northwest Institute of Eco-Environment and Resources, Chinese Academy of Sciences

zhangze@lzb.ac.cn

**Quntao Duan**

Northwest Institute of Eco-Environment and Resources, Chinese Academy of Sciences
[duanqt@lzb.ac.cn](mailto:duanqt@lzb.ac.cn)

**Deyu Tian**

Northwest Institute of Eco-Environment and Resources, Chinese Academy of Sciences
[tiandy@lzb.ac.cn](mailto:tiandy@lzb.ac.cn)

**Qingguo Zhou**

Lanzhou University
[zhouqg@lzu.edu.cn](mailto:zhouqg@lzu.edu.cn)

**Data Sources and Terms of Use**

The use of data is conditional on citing the original data sources. Full details on how to cite the data are given at the bottom of each page. For research projects, if the data are essential to the work, or if an important result or conclusion depends on the data, co-authorship may need to be considered. Permafrost engineering and slope monitoring facilitate the acquisition of data to encourage its use and promote understanding of the potential impact of freeze-thaw cycles on Permafrost engineering. Respecting original data sources is key to help secure the support of data providers to enhance, maintain and update valuable data.

**Acknowledgements**

Funded by the National Natural Science Foundation of China (41871065), the National Science Fund for Distinguished Young Scholars (41825015), the Key Research Project of Frontier Science of Chinese Academy of Sciences (QYZDJ-SSW-DQC040), and the Strategic Priority Research Program of the Chinese Academy of Sciences (XDA19090122).

**License**

Apache License 2.0

**Contact**

**Dr. Lihui Luo**
Northwest Institute of Eco-Environment and Resources, Chinese Academy of Sciences
[luolh@lzb.ac.cn](mailto:luolh@lzb.ac.cn)

updated: 2021/04/14

---

## Author Response (AR2)

**Authors reply to the comments by Jan Beutel of the manuscript essd-2020-106**

**"An integrated observation dataset of the hydrological-thermal deformation in permafrost slopes and engineering infrastructure in the Qinghai-Tibet Engineering Corridor"**

**by Lihui Luo et al.**

We thank Dr. Jan Beutel for valuable feedback, which helped us improve the manuscript. Please find below the Referee comments in black, Author responses in green, and Changes to the manuscript in blue.

**Response to referee comment 1:**

**Dear authors,**

as stated in the previous review report your submission is lacking w.r.t. the dataset preparation. I acknowledge your efforts done for this the revision, especially the preparation of an extensive README document as guidance, a working R codebase and the inclusion of the InSAR data, however data files mentioned are still missing and/or the data described cannot be found. Also, some of the files I downloaded from Zenodo are labeled as corrupt.

**Response:** Thank you for the insightful comments. We endeavored to improve the manuscript and datasets by your comments. To address these concerns, we have made the following modifications: (1) We have merged multiple meteorological or ground data files into one meteorological or ground data file, and it was updated to 2020 because a weather station contains multiple meteorological or ground data in different

years; (2) We have updated the README file to ensure that it is consistent with the naming in the data repository; and (3) we have added frozen soil classification and borehole figures, and added vector data for frozen soil classification in the Qinghai-Tibet Plateau. Because the Zenodo repository does not support directories, we have compressed some datasets. We guess that the data corruption may be caused by the decompression software of different operating systems. We have updated and checked every dataset in the data repository. We have no intention of concealing any marked data files. We are not only data producers, but also data users. Ensuring the integrity of data so that it can be used has always been our goal. If there is any data missing or corrupted, please feel free to contact us. For more details, please see our replies below.

In your response you add two figures R2/R3 as well as argumentation about certain subjects. If these figures/argumentation is important for the understanding of the manuscript they should be included there, and not only in the response of the authors. IMHO the figures R2 and R3 would be nice to have in the full paper.

**Response:** We agree. We have redrawn these two figures and put them in the appendix of the manuscript.

Appendix B: Drilling data source

[Figure]

Figure B1: The location of two drilled boreholes.

Appendix C: Classification of frozen soil

[Figure]

Figure C1: The frozen soil distribution in the study area.

Concerning data cleaning/validation you mention a column "corrected_p" and give a table 3 on correction values. however i cannot find these in any data file provided.

**Response:** Sorry, due to our mistake, the prepared data has not been uploaded to the data repository correctly. In fact, the "corrected_p" column is included in the data file that we uploaded before starting with "TRPFUPS" (DOI: 10.5281/zenodo.4588099). In order to solve the problem of multiple meteorological data files for a weather station, we merged multiple weather files from a weather station into one file.

You mention a multitude of data corrections as well as gap filling. The details remain unclear and especially w.r.t. figure 2 the utility is not given. you simply interpolate over large gaps of ~a year of data. That is not correct here. remove the green daily means at least for the large gaps.

**Response:** Gap fill is better for applications that lack a few data in the middle and have data before and after. It is unreliable for long-term lack of data. The quality control code 8 for meteorological data is used to fill in missing data. The missing data in Figure 2 mainly occurred from July 7, 2017 to October 3, 2017, which was caused by instrument failure. Filling in the data for this period is a difficult task. The drawing program in Figure 2 is in the 40-69 lines of the Meteorological.R code in "R code for permafrost indices and visualization". We use ggplot2 library to draw, and we need to convert multiple columns of data into single column data. The converted data has also been placed in the Data directory. The visualization of precipitation is a bar graph, and the other graphs are line graphs, so the precipitation data is visualized separately and then spliced with the previous graph. The files are XDTMS2014-2018_GT.csv and XDTMS2014-2018_PREC.csv. We have given instructions in the README file.

The README files you have created greatly simplify the data access. However the contain incorrect/incomplete information w.r.t. the data files provided. A number of the files listed in table 2 cannot be found in the repository or they are named slightly different. Furthermore the data variables listed in table 3 cannot be found. As such the

data description paper as well as the dataset are not in agreement and should be rejected.

**Response:** Sorry because of our incorrect naming and uploading. The data in Table 2 is divided into different directories according to different dates. In fact, the two data files (DOI: 10.5281/zenodo.4588099) Golmud_52818_2010_2018.xlsx and Wudaoliang_52908_2010_2018.xlsx contain part of the QC quality control in table 3. We merge multiple data files of each weather station into a ground and a weather csv or xlsx format data, and we checked and added all missing data files.

As a last comment i don't think it is necessary to make a copy of the readme file in each directory as well as in the supplement and it's subdirectories. rather than having many (copied) README file, concentrate on generating one that contains full and correct information.

**Response:** We kept a README file with complete information and updated this README file.

(In the README to your data)

Data Sources and Terms of Use

The use of data is conditional on citing the original data sources. Full details on how to cite the data are given at the bottom of each page. For research projects, if the data are essential to the work, or if an important result or conclusion depends on the data, co-authorship may need to be considered. Permafrost engineering and slope monitoring facilitate the acquisition of data to encourage its use and promote understanding of the potential impact of freeze-thaw cycles on Permafrost engineering. Respecting original data sources is key to help secure the support of data providers to enhance, maintain and update valuable data.

**Response:** All data in this manuscript follow the Apache License 2.0 (thanks to the editor for suggesting that we choose this license in the early stage of the manuscript), so we deleted this term in the README.

---

## Author Response (AR3)

**Authors reply to the comments by Editor Dr. Kirsten Elger and Dr. Jan Beutel of the manuscript essd-2020-106**

**"An integrated observation dataset of the hydrological-thermal deformation in permafrost slopes and engineering infrastructure in the Qinghai-Tibet Engineering Corridor"**

**by Lihui Luo et al.**

We thank Editor Dr. Kirsten Elger and Dr. Jan Beutel for valuable feedback, which helped us improve the manuscript and data quality. Please find below the Referee comments in black, Author responses in green, and Changes to the manuscript in blue.

**Response to Editor's comment:**

Comments to the Author:

Dear Lihui Luo and co-authors,

please address the new review report from 8 June 2021. It mostly addresses the some inconsistencies between the data and their description.

In addition, please cite the data from the Xidatan station that you have retrieved from the National Tibetan Plateau Data Center (https://doi.org/10.11888/Meteoro.tpdc.270084). For this, I suggest to change the following sentence (line 149/150 in the track change mode version) from

"Their data can be combined with the data obtained from the Xidatan field station to analyze the spatiotemporal dynamics of the permafrost slopes in the corridor"

to

"Their data can be combined with the data obtained from the Xidatan field station (Zhao, 2018) to analyze the spatiotemporal dynamics of the permafrost slopes in the corridor"

and add the following reference in the "references" section:

ZHAO Lin. Meteorological Datasets of Xidatan station (XDT) on the Tibetan Plateau in 2014-2018. National Tibetan Plateau Data Center. https://doi.org/10.11888/Meteoro.tpdc.270084, 2018.

Many thanks and best regards

Kirsten Elger

**Response:** Thank you for the insightful comments. We have cited the data and added the reference accordingly.

**Response to Referee's comment:**

Dear authors,

the last revision is greatly improved w.r.t. consistency and completeness. It is commendable that you also took the time to update some of the data up to the year 2020 (in part). However again here, consistency is key and consistency across the four data/code packages as well as the paper supplement you supply is not given. There are a few discrepancies left that you should/may address in order to facilitate re-use of this dataset:

- The README files given should be synchronized. The new README file in the supplement is different to the one in R-code/TLS data. Filenames, listings, license.

**Response:** Thank you for the positive and valuable feedback. As you recommended, we kept a README file with complete details and put the README file into each dataset. In addition, in the first TLS experiment, we also used Nikon D-series digital cameras to uninterruptedly photograph slope A along the Qinghai-Tibet Highway and finally generated point cloud data. This data has previously been put into the TLS data set, and we have added this information in the README file. Meanwhile, we have standardized the file names in the TLS data.

- The data files in the R-Code https://zenodo.org/record/4686141 are not in agreement with the data files in https://zenodo.org/record/4879639. Not concerning the data duration and especially not concerning the format. Therefore it is unclear to me which data is which? And what benefit the R-code really brings, apart from generating your two figures for the paper. It is certainly not a show killer, but it requires some manual checking by everyone that wants to build on your paper/data.

**Response:** Sorry for not updating the data in the R code, this time we put the latest data into the R code. The function of the R code is to calculate the two permafrost indices, MAAT and MAGST, in which the changes of the indices are closely related to the hydrological-thermal deformation of permafrost slopes. The variable name in the R

code is consistent with the variable name in the meteorological data. Only when users use their own data to calculate the permafrost indices, they need to change the variable name in the R code. In addition, we added an explanation in the R code.

- w.r.t. figure 2 my gap filling comment earlier was specifically geared at the one year+ gap on the wind data for 2014-2016. Here the green line should really removed as the wind speed is not deviating that much. Also it is a pity you supply data up to recently (2020) but do not show this data in one of the plots.

**Response:** Thank you for reminding us to pay attention to this problem again. We removed the green lines. As recommended, we have redrawn Figure H1 so that the data in the figure is updated to 2020. Meanwhile, we updated the R code and corresponding data.

- The data files in https://zenodo.org/record/4879639 have different formats. While the data in the .CSV is explained in detail (README, tables, columns, quality control, corrections) Meteo_00000_Xidatan_2014-2019.xlsx and GT_00000_Slopes_2014-2019.xlsx are not explained. Specifically, if these files are not quality controlled/checked this should be specifically mentioned.

**Response:** We converted Meteo_00000_Xidatan_2014-2019.xlsx to Meteo_XDTMS_Xidatan_2014-2018.csv. Another sheet of this file contains the description information of the data, and this station has an internal ID (XDTMS) and it is mentioned in Table 1. We added the data description of the Xidatan and Slope in REAME.md file. Regarding quality control, we have added the following sentence:

However, the meteorological data of Xidatan field station and the ground data of the study area are manually sorted and verified, and no standardized quality control is adopted.

- the xidatan Meteo data already has a DOI citation 10.11888/Meteoro.tpdc.270084. this one should be specified in full as it allows the reader to obtain regular updates

autonomously.

**Response:** Thank you for the insightful comments. We have cited the data and added the reference accordingly.

\- It is great that your UAV/RBG/TIR data now works.

**Response:** Thank you for the positive and valuable feedback.